# Embodied Interpretability: Linking Causal Understanding to Generalization in Vision-Language-Action Models

Hanxin Zhang [1 2]   Mingshuo Xu [1 2]   Abdulqader Dhafer [1 2]   Shigang Yue [2]   Hongbiao Dong [3]   Zhou Daniel Hao [1 2]

 https://github.com/robot-future/vla-explain

## Abstract

Vision–Language–Action (VLA) policies often fail under distribution shift, suggesting that decisions may depend on spurious visual correlations rather than task-relevant causes. We formulate visual–action attribution as an interventional estimation problem. Accordingly, we introduce the **Interventional Significance Score (ISS)**, an interventional masking procedure for estimating the causal influence of visual regions on action predictions, and the **Nuisance Mass Ratio (NMR)**, a scalar measure of attribution to task-irrelevant features. We analyze the statistical properties of ISS and show that it admits unbiased estimation, and we characterize conditions under which action prediction error provides a valid proxy for causal influence. Experiments across diverse manipulation tasks indicate that NMR predicts generalization behavior and that ISS yields more faithful explanations than existing interpretability methods. These results suggest that interventional attribution provides a simple diagnostic approach for identifying causal misalignment in embodied policies.

## 1. Introduction

Vision-Language-Action (VLA) models (Zhen et al., 2024; Chen et al., 2025; Zhang et al., 2025d;c; Huang et al., 2025; Shi et al., 2025) have demonstrated significant capabilities in embodied intelligence. However, our empirical analysis reveals two critical anomalies in current mainstream research: (1) attention scores predominantly activate on

[1]DANiLab, University of Leicester, UK [2]School of Computing and Mathematical Sciences, University of Leicester, UK [3]School of Metallurgy and Materials, University of Birmingham, UK . Correspondence to: Zhou Daniel Hao <d.hao@leicester.ac.uk>.

*Proceedings of the 43rd International Conference on Machine Learning*, Seoul, South Korea. PMLR 306, 2026. Copyright 2026 by the author(s).

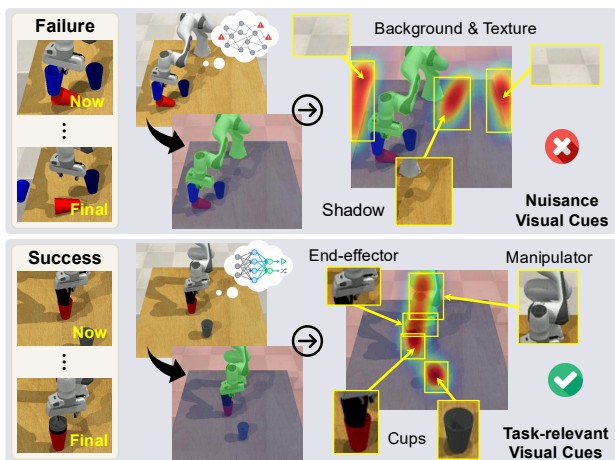

*Figure 1.* **Generalization of VLAs can be analyzed through action attribution.** For instance, in the task *"stack the other cups on the top of the red cup"*, failed trials rely more on nuisance visual cues (e.g., background, texture, and shadows) for decisions; successful trials rely more on task-relevant cues (e.g., manipulator, end-effector, and cups).

task-irrelevant regions, such as the background (Kachaev et al., 2025; Zhong et al., 2025; Gong et al., 2025); and (2) the output actions continue to follow a similar trend even when the visual inputs are completely masked (Omaisan & Mohamed, 2025; Fei et al., 2025; Kim et al., 2025). These phenomena suggest that VLA models may rely on memorizing statistical mappings between tasks and actions rather than learning the underlying causal mechanisms. We hypothesize that this reliance on spurious correlations (Zhang et al., 2025e) is a primary factor contributing to their limited Out-of-Distribution (OOD) generalization, as shown in Figure 1. This raises a fundamental question: How can we quantify whether a VLA policy relies on causally relevant visual evidence rather than spurious correlations, and does this reliance predict generalization performance?

While existing interpretability research in embodied intelligence (Häon et al., 2025; Peng et al., 2025; Zheng et al., 2025; Wu et al., 2025) has not fully resolved this challenge, it offers valuable analytical priors. For instance, visualizing attention weights identifies focal regions, and inserting

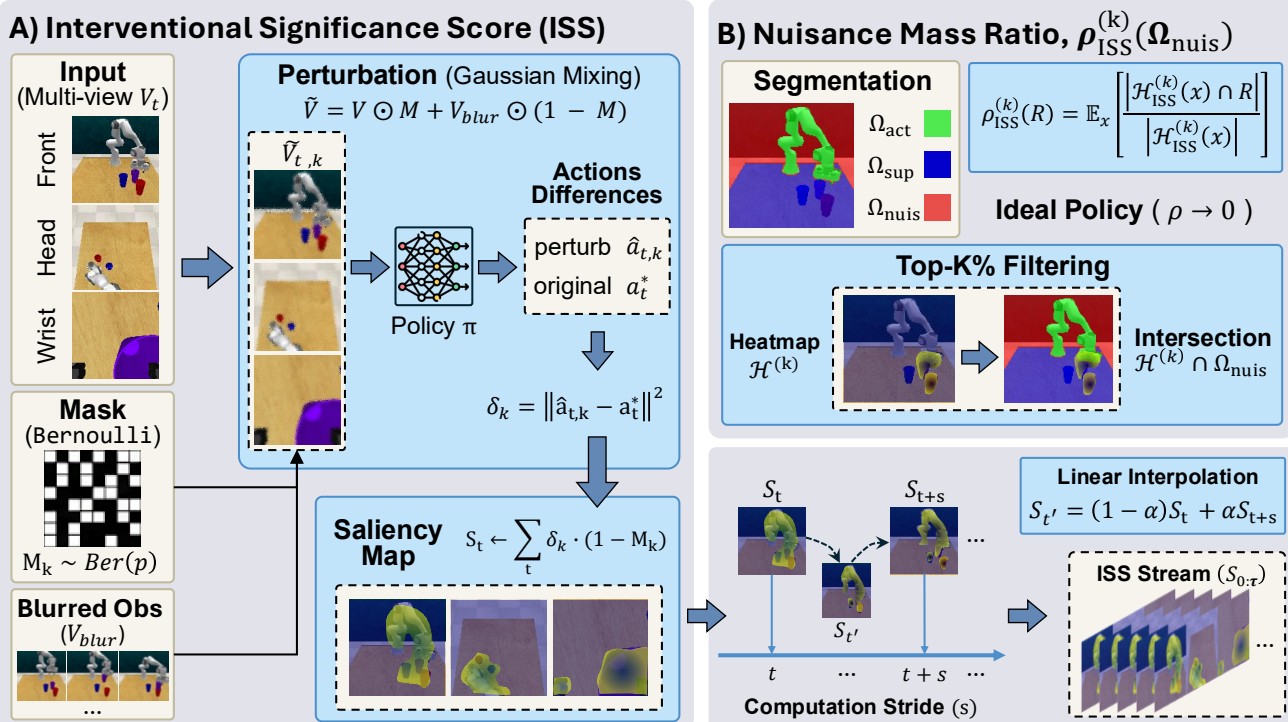

*Figure 2.* **The overview of the proposed interpretability approach.** Panel (A) illustrates the pipeline for generating the Interventional Significance Score (ISS), where action discrepancies ($\delta_k$) induced by Bernoulli masking ($M_k$) and Gaussian mixing perturbations ($V_{blur}$) on multi-view observations are aggregated to yield saliency maps, subsequently forming a continuous ISS stream via linear interpolation. Panel (B) defines the Nuisance Mass Ratio (NMR) metric, which quantifies policy reliance on non-causal features by computing the normalized intersection between the top-k filtered saliency regions ($\mathcal{H}^{(k)}$) and predefined nuisance segments ($\Omega_{nuis}$).

probes into intermediate hidden layers confirms the encoding of semantic concepts. These findings substantiate the utility of interpretability tools for diagnosing model behaviors. Building on this, we investigate whether integrating mechanistic interpretability with interventional causality effectively diagnoses generalization by evaluating the extent to which the model captures causal relationships.

In this work, we introduce a causal testing method that assesses whether robot policies rely on the correct visual regions and empirically show that this measure predicts how well they generalize to new environments. The key contributions are as follows:

- We formulate visual–action attribution as an interventional estimation problem and introduce two measures: the Interventional Significance Score (ISS), which estimates the causal influence of visual regions on action decisions, and the Nuisance Mass Ratio (NMR), which quantifies attribution to task-irrelevant features. (Section 3, Section 4)

- We demonstrate empirically that NMR is predictive of out-of-distribution generalization, and that ISS produces more consistent and informative attributions than

existing interpretability methods across diverse manipulation tasks. (Section 5, Appendix D)

- We provide theoretical analysis showing that ISS admits unbiased estimation and characterize conditions under which action prediction error serves as a valid surrogate for distributional divergence, establishing a principled basis for interpreting the proposed measures. (Appendix A)

## 2. Related Work

While Vision-Language-Action (VLA) models have demonstrated remarkable capabilities in end-to-end control, their internal decision-making mechanisms remain largely opaque "black boxes" (Zhang et al., 2025b). Initial efforts to address this sought transparency through modular designs or explicit reasoning. For instance, SayCan (Ichter et al., 2023) and Code-as-Policies (Ahn et al., 2025) decoupled planning from execution by leveraging the probabilistic outputs or code generation capabilities of Large Language Models. Similarly, VoxPoser (Huang et al., 2023) visualized abstract instructions by generating 3D value maps. To enhance the readability of the reasoning process, recent works have incorporated Chain-of-Thought mechanisms:

CoT-VLA (Zhao et al., 2025) autoregressively generates visual subgoals, PhysiAgent (Wang et al., 2025) introduces self-reflection modules, and CoC-VLA (Zhang et al., 2025a) constructs causal chains to explain decisions in autonomous driving. However, these approaches predominantly focus on system-level transparency or the rationality of intermediate outputs rather than on uncovering feature-level operating mechanisms. Consequently, recent research has shifted towards mechanistic interpretability, clustering primarily around attention analysis, latent state probing, and feature disentanglement.

**Attention Analysis.** Analyzing attention weights serves as a direct method for localizing regions of interest. Mitra et al. (2025) proposed "Robotic Steering," which identifies and fine-tunes task-specific attention heads using few-shot demonstrations, thereby correcting behavior without altering global model parameters. To address visual redundancy, the DTP framework (Li et al., 2026) uses attention heatmaps to identify and prune visual tokens that distract from decision-making. Although they indicate where a model focuses, they do not establish whether those regions are causally necessary for the resulting actions. Consequently, it remains unclear how to quantitatively assess whether a VLA policy truly depends on task-relevant evidence or on nuisance features. This motivates the central question of this work: can we quantify the extent to which a VLA policy's actions depend on causally relevant versus nuisance visual evidence, and does this dependence predict generalization performance?

**Latent State Probing.** Probing techniques are widely employed to decipher the semantic content of hidden representations. Lu et al. (2025) demonstrated, using linear probes, that OpenVLA's hidden layers explicitly encode symbolic states, such as object attributes and spatial relations. Going further, Molinari et al. (2025) used embedding arithmetic to demonstrate the emergence of internal world models within VLAs that can predict environmental dynamics. In the domain of motion planning, Tas & Wagner (2025) showed how to extract control vectors from Motion Transformers (Nayakanti et al., 2023) to interpret motion parameters. Additionally, the CURE framework by Yin et al. (2025) quantifies epistemic uncertainty by analyzing hidden layer features to assess task familiarity. However, a fundamental flaw of probing techniques lies in their passivity; they rely on supervised training with predefined concept sets. Furthermore, proving that information "exists" does not imply that the model actually "uses" it for decision-making, leading to a causal disconnect between observation and control logic.

**Feature Disentanglement.** To achieve fine-grained control, researchers have begun decomposing dense neural activations into sparse, interpretable features. Häon et al. (2025) introduced a method that projects Feed-Forward Network (FFN) vectors into token space and clusters them, enabling semantic steering of VLA behavior at inference time. In the realm of safety, Wen et al. (2025) introduced concept-based dictionary learning, which uses Singular Value Decomposition (SVD)-based Principal Component Analysis (PCA) to extract directions representing unsafe behaviors from activations. While sparse decomposition offers a pathway to disentangle polysemanticity, existing methods struggle with complex, compositional embodied concepts, and the computational overhead of training large-scale autoencoders limits their potential for real-time application.

## 3. Preliminary

We formalize the problem setting by defining Vision-Language-Action models and reviewing the probabilistic machinery of Markov Blankets necessary to distinguish causal mechanisms from redundant statistical dependencies.

### 3.1. Vision-Language-Action Models

Vision-Language-Action (VLA) models (Brohan et al., 2023; Zitkovich et al., 2023; Kim et al., 2024; Mees et al., 2024; Black et al., 2025; Zhou et al., 2025b) operate as a learned policy $\pi_\theta$ parameterized by $\theta$, mapping multimodal observations to actions. Formally, the policy predicts an action $\mathbf{a}_t$ at time step $t$ conditioned on a multimodal context $\mathbf{X}_t$:

$$\mathbf{a}_t \sim \pi_\theta(\cdot \mid \mathbf{X}_t) \tag{1}$$

The context $\mathbf{X}_t$ is constructed by concatenating the sequence of visual patch embeddings $(\mathbf{v}_1, \ldots, \mathbf{v}_{n_1}) \in \Omega$, and the linguistic instruction tokens $(\mathbf{l}_1, \ldots, \mathbf{l}_{n_2}) \in O$:

$$\mathbf{X}_t = [\mathbf{v}_1, \ldots, \mathbf{v}_{n_1}, \mathbf{l}_1, \ldots, \mathbf{l}_{n_2}] \in \mathbb{R}^{(n_1+n_2) \times d}, \tag{2}$$

where $n_1$ and $n_2$ denote the sequence lengths of visual patches and instruction tokens, respectively, and $d$ is the embedding dimension.

### 3.2. Conditional Independence and Markov Blankets

To formalize the causal dependencies between observations and actions under instruction, we utilize the concept of conditional independence. Let $X, Y, Z$ be random variables. We say $X$ is conditionally independent of $Y$ given $Z$, denoted as $X \perp\!\!\!\perp Y \mid Z$, if $P(X \mid Y, Z) = P(X \mid Z)$.

In the context of graphical models, the **Markov Blanket** (Pearl, 1988; Verma & Pearl, 2022) of a target variable $T$, denoted as $\mathcal{M}(T)$, is the minimal subset of variables such that $T$ is conditionally independent of all other variables in the system $\mathbf{F}$ excluding $T$ given $\mathcal{M}(T)$. Formally, for a feature set $\mathbf{F}$:

$$T \perp\!\!\!\perp (\mathbf{F} \setminus \mathcal{M}(T)) \mid \mathcal{M}(T). \tag{3}$$

Intuitively, $\mathcal{M}(T)$ contains all the information sufficient to infer $T$. In this work, we extend this notion to identify the minimal regions in $\Omega$ required to recover the expert action $\mathbf{a}$.

# 4. Theoretical Framework

We introduce a causal analysis approach that quantifies causal effects via intervention and then diagnoses causal misalignment in Vision-Language-Action (VLA) models. The overview is demonstrated in Figure 2.

## 4.1. Causality Quantification within Latent Space

To quantify the causal necessity of specific input features, we propose a method for quantifying causality in latent space that measures counterfactual information divergence. The proposed quantification method follows a *do*-calculus paradigm, using a mean-based replacement in interventions and an accelerated approach in calculus.

### 4.1.1. CAUSAL INTERVENTION

To measure the causal contribution of the $i$-th token of $\mathbf{X}_t$, we approximate the marginalization of information by replacing the token with a static baseline derived from the training distribution $\mathcal{D}$. To preserve distributional validity across modalities, we define specific baselines for visual and linguistic inputs:

$$\tilde{\mathbf{X}}_t^{(i)} = [\mathbf{x}_1, \ldots, \mathbf{x}_{i-1}, \boldsymbol{\mu}_i, \mathbf{x}_{i+1}, \ldots, \mathbf{x}_{n_1+n_2}] \quad (4)$$

where $\mathbf{x}_i \in \mathbf{X}_t$, defined in Eq. (2), and $\boldsymbol{\mu}_i$ is the marginal mean embedding conditioned on the modality of token $i$:

$$\boldsymbol{\mu}_i = \begin{cases} \mathbb{E}_{\mathbf{v} \sim \mathcal{D}_{\text{vis}}}[\mathbf{v}] & \text{if } i \leq n_1 \quad \text{(Visual Mean)} \\ \mathbb{E}_{\mathbf{l} \sim \mathcal{D}_{\text{text}}}[\mathbf{l}] & \text{if } i > n_1 \quad \text{(Textual Mean)} \end{cases} \quad (5)$$

This formulation constrains the ablated sequence $\tilde{\mathbf{X}}_t^{(i)}$ to the valid semantic subspace of the respective modality, effectively mitigating the out-of-distribution (OOD) artifacts associated with conventional zero-ablation.

### 4.1.2. INTERVENTIONAL SIGNIFICANCE SCORE (ISS)

With the counterfactual policy introduced above and the full-information policy, causality can be reflected by the Interventional Significance Score (ISS) defined via divergence. To isolate the causal effect of input tokens on the current decision step without compounding errors from trajectory divergence, we evaluate ISS under teacher forcing (Williams & Zipser, 1989), conditioning both distributions on the ground-truth action $\pi_\theta(\cdot \mid \mathbf{X}_t)$.

**Definition 4.1** (Interventional Significance Score). The causal significance of token $i$ is quantified by the expected cumulative Kullback-Leibler (KL) divergence between the original distribution and the interventional distribution, conditioned on the expert trajectory:

$$\text{ISS}_i(\theta) = \mathbb{E}_{\mathbf{X} \sim \mathcal{D}} \left[ \sum_{t=1}^{T} D_{\text{KL}} \left( \pi_\theta(\cdot \mid \mathbf{X}_t) \,\|\, \pi_\theta(\cdot \mid \tilde{\mathbf{X}}_t^{(i)}) \right) \right] \quad (6)$$

Direct computation of Eq. (6) is intractable for high-dimensional inputs. To address this, we employ a Monte Carlo estimation strategy using randomized binary masks (Algorithm 1). As shown in Appendix A.1, this stochastic method is a consistent estimator of coalitional causal effects. Furthermore, under the fixed isotropic Gaussian policy assumption adopted in VLA training, the Fisher Information Matrix with respect to the action mean parameters reduces to a scaled identity matrix, implying that the KL divergence is proportional to the squared difference between predicted action means. Consequently, we use Action MSE as a computationally efficient proxy for distributional divergence in our implementation, justified by the closed-form equivalence between KL divergence and squared action-mean differences under a fixed isotropic Gaussian policy, as detailed in Appendix A.2.

## 4.2. Causal Misalignment in Learned Subspaces

In this section, we propose an approach for spatial causal attribution. We begin by partitioning the observation space based on conditional independence properties. Building on this topology, we derive a metric to quantify *causal misalignment*, the extent to which the policy's sensitivity leaks into the irrelevant nuisance subspace.

### 4.2.1. CAUSAL SPATIAL PARTITION

We consider a sequential decision-making setting or imitation learning setting defined by an observation space $\Omega$ (derived from the underlying state space $\mathcal{S}$), an action space $\mathcal{A}$, and an instruction space $\mathbf{L}$. Let $\pi^* : \Omega \xrightarrow{L} \Delta(\mathcal{A})$ denote the expert policy that maps observations $\mathbf{v} \in \Omega$ to a distribution over actions $\mathbf{a} \in \mathcal{A}$. To analyze the causal role of different image regions, we decompose $\Omega$ into distinct subsets relative to the task instruction $L$ and the ideal expert policy $\pi^*$.

**Definition 4.2** (Causal Spatial Partition). The token space $\Omega$ is partitioned into three disjoint sets $\Omega = \Omega_{act} \cup \Omega_{sup} \cup \Omega_{nuis}$:

1. Action-Critical Regions ($\Omega_{act}$): Tokens representing the robotic embodiment itself, specifically the manipulator arm and the end-effector.

2. Environmental Support Regions ($\Omega_{sup}$): Tokens representing task-relevant objects and essential physical

**Algorithm 1** Interventional Significance Score (ISS)

1: **Input:** VLA Policy $\pi_\theta$, Visual Sequence $\mathbf{V}_{1:T}$, Number of input tokens $M = n_1 + n_2$, Number of Masks $N$, Keep Probability $p$, Temporal Stride $s$, Blur Kernel $\mathcal{K}_\sigma$
2: **Output:** Saliency Maps $\mathbf{S}_{1:T}$
3: **Complexity:** Time $\mathcal{O}(\lceil \frac{T}{s} \rceil \cdot N \cdot C_\pi)$, Space $\mathcal{O}(T \cdot H \cdot W)$
4: Initialize $\mathbf{S}_{1:T} \leftarrow \mathbf{0}$
5: Generate binary masks $\mathbf{m} \in \{0,1\}^M$ where $\mathbf{m}_k \sim$ Bernoulli$(p)$
6: Pre-compute blurred sequence $\mathbf{V}_t^{blur} \leftarrow \mathbf{V}_t * \mathcal{K}_\sigma$ for $t \in \{1, \ldots, T\}$
7: **for** $t = 1$ **to** $T$ **step** $s$ **do**
8:     $a_t^* \leftarrow \pi_\theta(\mathbf{V}_t)$
9:     $\delta \leftarrow \mathbf{0}$
10:     **for** $k = 1$ **to** $N$ **do**
11:         $\tilde{\mathbf{V}}_{t,k} \leftarrow \mathbf{V}_t \odot \mathbf{m}_k + \mathbf{V}_t^{blur} \odot (\mathbf{1} - \mathbf{m}_k)$
12:         $\hat{a}_{t,k} \leftarrow \pi_\theta(\tilde{\mathbf{V}}_{t,k})$
13:         $\delta_k \leftarrow \|\hat{a}_{t,k} - a_t^*\|_2^2$
14:         $\mathbf{S}_t \leftarrow \mathbf{S}_t + \delta_k \cdot (\mathbf{1} - \mathbf{m}_k)$
15:     **end for**
16:     $\mathbf{S}_t \leftarrow \mathbf{S}_t \oslash (N \cdot (1 - p))$
17: **end for**
18: **for** $t = 1$ **to** $T$ **do**
19:     **if** $t \pmod s \neq 1$ **then**
20:         $t_{prev} \leftarrow s \cdot \lfloor \frac{t-1}{s} \rfloor + 1$
21:         $t_{next} \leftarrow \min(t_{prev} + s, T)$
22:         $\alpha \leftarrow (t - t_{prev}) / (t_{next} - t_{prev})$
23:         $\mathbf{S}_t \leftarrow (1 - \alpha) \cdot \mathbf{S}_{t_{prev}} + \alpha \cdot \mathbf{S}_{t_{next}}$
24:     **end if**
25: **end for**
26: **return** $\mathbf{S}_{1:T}$

support structures, such as table surfaces or racks required for execution.

3. Visual Nuisance Regions ($\Omega_{nuis}$): Tokens representing elements that do not physically interact with the task, such as background wall colors, lighting reflections, or distractor objects.

Crucially, the definition above pertains to the latent token space $\Omega$ rather than the raw pixel space. While pixel-level representations often suffer from entanglement where a change in lighting affects all pixels, the token space provides a higher-level semantic abstraction. In this latent space, semantic entities are spatially disentangled, making the separation between causal elements and nuisances distinct.

This structural characteristic of the token space allows us to derive the following theoretical properties regarding the expert policy.

**Proposition 4.3.** *The expert policy $\pi^*$ satisfies the condi-*

*tional independence property:*

$$\pi^*(\mathbf{a} \mid \mathbf{X}_\Omega) = \pi^*(\mathbf{a} \mid \mathbf{X}_{\Omega_{act}}, \mathbf{X}_{\Omega_{sup}}) \quad (7)$$

*where $\mathbf{X}_\Omega$ denotes the subset of input features corresponding to the region $\Omega$.*

**Proposition 4.4** (Causal Markov Blanket). *The union of the action-critical and support regions constitutes the Causal Markov Blanket for the action variable $\mathbf{a}$:*

$$\mathcal{M}(\mathbf{a}) = \Omega_{act} \cup \Omega_{sup} \quad (8)$$

Theoretical Rationale: This formulation applies the concept of a Markov Blanket from Bayesian networks. It asserts that $\Omega_{act}$ and $\Omega_{sup}$ contain all the necessary information to determine the optimal action $a$. Consequently, any statistical dependence of a learned model $\pi_\theta$ on $\Omega_{nuis}$ constitutes a causal hallucination, indicating that the model is relying on spurious correlations rather than physical causality.

#### 4.2.2. THE CAUSAL MISALIGNMENT METRIC

Based on the spatial partition, we define a quantitative metric to evaluate the density of critical tokens within any given region $R$ under an importance scoring function $\phi$.

**Definition 4.5** (Regional Mass Ratio). Let $\phi(\mathbf{X})$ be the non-negative importance score (e.g., Attention Score or Interventional Significance) for given input $\mathbf{X}$. Let $\mathcal{H}_\phi^{(k)}(\mathbf{X})$ denote the set of critical tokens accounting for the top $k\%$ of the cumulative importance mass based on $\phi$. For a specific spatial region $R \in \{\Omega_{act}, \Omega_{sup}, \Omega_{nuis}\}$, the Regional Mass Ratio is defined as:

$$\rho_\phi^{(k)}(R) = \mathbb{E}_{\mathbf{X}} \left[ \frac{\left| \mathcal{H}_\phi^{(k)}(\mathbf{X}) \cap R \right|}{\left| \mathcal{H}_\phi^{(k)}(\mathbf{X}) \right|} \right] \quad (9)$$

While the Regional Mass Ratio provides a general framework for quantifying token distribution across any region, detecting causal misalignment requires focusing specifically on the irrelevant background. Therefore, we instantiate this metric for the nuisance region $\Omega_{nuis}$ using the Interventional Significance Score (ISS), termed the Nuisance Mass Ratio (NMR):

**Definition 4.6** (Nuisance Mass Ratio (nmr@k)). The Nuisance Mass Ratio is defined as:

$$\rho_{\text{ISS}}^{(k)}(\Omega_{nuis}) = \mathbb{E}_{\mathbf{X}} \left[ \frac{\left| \mathcal{H}_{\text{ISS}}^{(k)}(\mathbf{X}) \cap \Omega_{nuis} \right|}{\left| \mathcal{H}_{\text{ISS}}^{(k)}(\mathbf{X}) \right|} \right] \quad (10)$$

The rationale and physical meaning of nmr@k are detailed below:

- Metric Meaning: This specifically quantifies Causal Misalignment. It measures the proportion of tokens identified as "causally significant" (via ISS) that actually belong to the irrelevant background ($\Omega_{nuis}$).

- Physical Scenario: In a robotic pick-and-place task, if the model erroneously relies on a specific wall texture or a light reflection to predict the gripper's movement, these background tokens will exhibit high ISS values. Consequently, $\rho_{\text{ISS}}^{(k)}(\Omega_{nuis})$ will be high, indicating that the policy is overfitting to spurious correlations.

- Ideal Behavior: A robust expert policy should satisfy $\rho_{\text{ISS}}^{(k)}(\Omega_{nuis}) \approx 0$, implying that its decision-making is entirely driven by the agent and relevant objects ($\Omega_{act} \cup \Omega_{sup}$), effectively ignoring the visual nuisances.

# 5. Experiments & Results

We empirically validate the proposed framework on the AG-NOSTOS benchmark (Zhou et al., 2025a) using a strictly offline interventional protocol to decouple causal mechanisms from simulator artifacts. All experiments are conducted on a VLA policy $\pi_{0.5}$ (Zhou et al., 2025b). We use 3600 episodes from the seen tasks $S$ for supervised fine-tuning $\pi_{0.5}$, and 575 episodes from the unseen tasks $U$ for evaluation. The unseen tasks $U$ are divided into two levels of generalization, denoted as $U_1$ and $U_2$. Subset $U_1$ consists of 13 tasks that share partial semantic overlap with $S$, such as similar object categories like cups or comparable action primitives such as putting. In contrast, subset $U_2$ contains 10 tasks that introduce entirely novel scenarios, with no overlapping objects or actions relative to $S$. More training details and results about VLA policy $\pi_{0.5}$ can be found in Appendix C.

## 5.1. Task Success Rate and NMR at Top-$k$

We compute the Pearson correlation coefficient ($r$) between the Nuisance Mass Ratio (nmr@k) and task success rate. More specifically, we evaluated the success rate of $\pi_{0.5}$ using the RLBench simulator (James et al., 2020) across 5 random seeds. For each seed, we evaluated all 41 tasks in the seen and unseen sets ($S$, $U_1$, and $U_2$) across 25 trials and computed their mean success rate and the Interventional Significance Score (ISS). We assessed Pearson correlation coefficients ($r$) across $k \in \{1, 5, 10, 15, 20\}$; Figure 3 highlights the optimal NMR@10 scenario yielding a maximal negative correlation of $-0.77$, while full results are detailed in Appendix D.1.

## 5.2. Robustness and Fidelity of ISS

Following the intervention semantics in structural causal models (Pearl, 2009; 2012), we design two types of experiments to evaluate explanatory robustness and fidelity. We

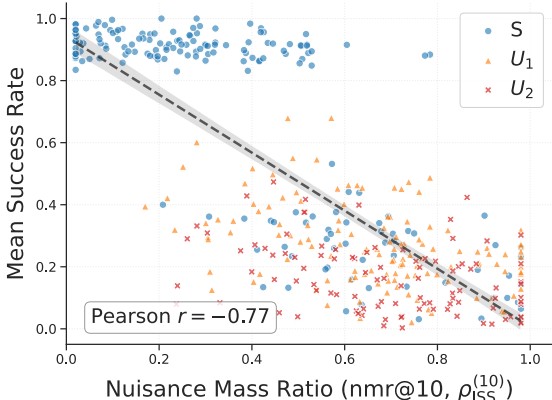

*Figure 3.* **Relationship between nmr@k and success rate.** Each sample point corresponds to the success rate of a single task evaluated under one random seed. For each of the 5 random seeds, 41 different tasks were evaluated, with each task executed over 25 trials to compute its success rate.

adopt **soft interventions** to assess robustness by injecting Gaussian noise into nuisance regions of the visual inputs, which increases input variability while preserving the underlying causal mechanisms. In contrast, we adopt **hard interventions** to assess fidelity by explicitly modifying nuisance regions via three controlled perturbations: textural, geometric, and patch-based. To benchmark our approach, we compare ISS with attention score (ATT) and token norm (NORM), and provide details on the processing of these interpretability baselines in Appendix E.

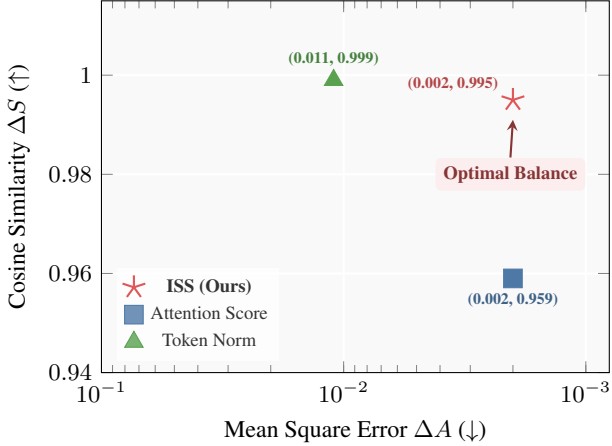

*Figure 4.* **Optimal Robustness.** ISS (red star) occupies the optimal top-right region, simultaneously maximizing the cosine similarity of the saliency map and minimizing the action's MSE compared to other explanatory methods.

**Robustness study.** We add Gaussian noise to the full-image input and select the top 5% episodes with the lowest action Mean Squared Error (MSE) across all tasks, yielding 200 episodes. Inspired by prior robustness studies (Cohen et al., 2019; Salman et al., 2019), we normalize the visual inputs

and set the Gaussian noise standard deviation to 0.25 in pixel space, aiming to increase input entropy without disrupting semantic structure. During the experiments, we inject noise only into nuisance regions and evaluate saliency stability using cosine similarity ($\Delta S$) and action MSE ($\Delta A$) between clean and perturbed maps. We report episode-averaged cosine similarity and action MSE in Figure 4. This Pareto plot details cosine similarity on the y-axis and action MSE on the x-axis, positioning ISS at $(0.002, 0.995)$, attention-based saliency at $(0.002, 0.959)$, and token-norm saliency at $(0.011, 0.999)$. Crucially, to align with the robustness objective of minimizing action deviation ($\Delta A \downarrow$) while maximizing saliency consistency ($\Delta S \uparrow$), we invert the x-axis to situate the optimal trade-off in the top-right quadrant.

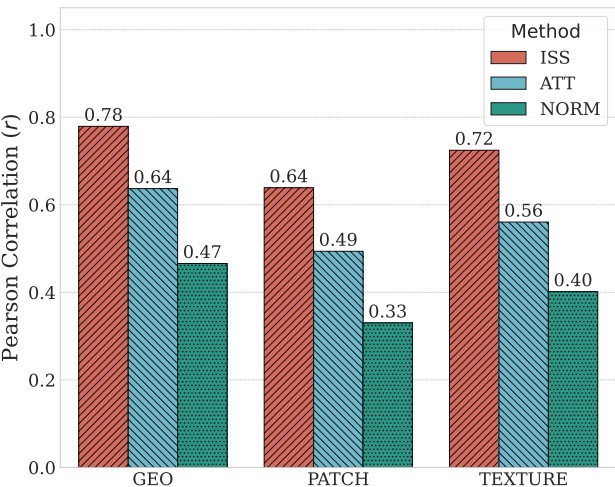

*Figure 5.* **Saliency Fidelity Analysis.** The bar chart displays the correlation between action MSE and saliency map changes across geometric, patch, and texture perturbations. ISS consistently shows a stronger linear alignment and higher Pearson coefficients than the ATT and NORM baselines.

**Fidelity study.** We apply three types of structured perturbations to nuisance regions across 1,000 randomly sampled episodes, including texture, geometric, and patch perturbations. For each episode, we compute saliency maps for both the clean input $S$ and the perturbed input $S'$ using ISS. For analysis, we continue to use cosine similarity between $S$ and $S'$ and examine its relationship with the action MSE $\Delta A$. Figure 5 summarizes the mean results under the three types of perturbations, comparing ISS with attention scores (ATT) and token norm (NORM). The results show Pearson correlation coefficients of 0.78, 0.64, and 0.72 for ISS across the three perturbation settings. Correspondingly, ATT yields 0.64, 0.49, and 0.56, while NORM yields 0.47, 0.33, and 0.40. We provide the detailed results in Appendix D.1.

### 5.3. Hyperparameter of ISS

We evaluate the sensitivity of ISS to the number of Monte Carlo interventions N and masking probability p, sweeping

*Table 1.* **Hyperparameter Analysis of ISS.** We evaluate the intervention count $N$ and masking ratio $p$ by measuring the average Action MSE between original and masked ISS across 500 sampled episodes (5 seeds). The configuration ($N = 100, p = 0.3$) is selected as optimal for yielding minimal action perturbation.

| Interventions ($N$) | Mask Ratio ($p$) | Seen MSE ($\times 10^{-3}$) | Unseen MSE ($\times 10^{-3}$) | Avg. MSE ($\times 10^{-3}$) |
|---|---|---|---|---|
| | 0.1 | $1.2 \pm 0.1$ | $8.5 \pm 0.3$ | $4.9 \pm 0.2$ |
| | 0.2 | $1.1 \pm 0.1$ | $9.0 \pm 0.4$ | $5.1 \pm 0.2$ |
| 50 | 0.3 | $1.5 \pm 0.2$ | $9.5 \pm 0.5$ | $5.5 \pm 0.3$ |
| | 0.4 | $1.2 \pm 0.1$ | $8.8 \pm 0.3$ | $5.0 \pm 0.2$ |
| | 0.5 | $1.4 \pm 0.2$ | $9.2 \pm 0.4$ | $5.3 \pm 0.3$ |
| | 0.1 | $1.2 \pm 0.1$ | $7.5 \pm 0.2$ | $4.4 \pm 0.1$ |
| | 0.2 | $1.0 \pm 0.0$ | $6.8 \pm 0.2$ | $3.9 \pm 0.1$ |
| **100** | **0.3** | $\mathbf{1.0 \pm 0.1}$ | $\mathbf{6.4 \pm 0.2}$ | $\mathbf{3.7 \pm 0.1}$ |
| | 0.4 | $1.1 \pm 0.1$ | $6.9 \pm 0.2$ | $4.0 \pm 0.1$ |
| | 0.5 | $1.2 \pm 0.1$ | $7.5 \pm 0.3$ | $4.4 \pm 0.2$ |
| | 0.1 | $1.5 \pm 0.2$ | $8.0 \pm 0.3$ | $4.8 \pm 0.2$ |
| | 0.2 | $1.4 \pm 0.1$ | $7.5 \pm 0.2$ | $4.5 \pm 0.1$ |
| 150 | 0.3 | $1.2 \pm 0.1$ | $7.0 \pm 0.2$ | $4.1 \pm 0.1$ |
| | 0.4 | $1.8 \pm 0.2$ | $8.5 \pm 0.4$ | $5.2 \pm 0.3$ |
| | 0.5 | $2.0 \pm 0.3$ | $9.0 \pm 0.5$ | $5.5 \pm 0.4$ |
| | 0.1 | $1.5 \pm 0.2$ | $9.5 \pm 0.4$ | $5.5 \pm 0.3$ |
| | 0.2 | $1.2 \pm 0.1$ | $10.0 \pm 0.5$ | $5.6 \pm 0.3$ |
| 200 | 0.3 | $1.5 \pm 0.2$ | $9.0 \pm 0.4$ | $5.3 \pm 0.3$ |
| | 0.4 | $1.2 \pm 0.1$ | $11.0 \pm 0.6$ | $6.1 \pm 0.3$ |
| | 0.5 | $1.1 \pm 0.1$ | $12.0 \pm 0.8$ | $6.6 \pm 0.4$ |

the parameter space of N $\in [50, 200]$ with a stride of 50 and p $\in [0.1, 0.5]$ with a stride of 0.1. Across 5 random seeds, we conducted a quantitative analysis by sampling 500 episodes per configuration to compute the Action MSE for each episode. We aim to select the ISS configuration that minimizes the interventional Action MSE on the original episodes.

As shown in Tab. 1, the quantitative comparisons indicate that increasing the masking ratio to p = 0.5 consistently results in elevated mean errors and standard deviations across all N settings (e.g., reaching $5.5 \times 10^{-3} \pm 0.4$ at N = 150). In contrast, the configuration (N = 100, p = 0.3) records the lowest error magnitude and variance, achieving a global minimum Average MSE of $3.7 \times 10^{-3} \pm 0.1$. We employed this ISS configuration (N = 100, p = 0.3) for all other experiments reported throughout the paper.

### 5.4. Additional Results

**Computation Efficiency.** We present total wall-clock latency, including CPU mask generation and batched GPU inference, relative to single inference (0.079s or 12.7Hz) across N $\in [50, 200]$ with a stride of 50 and p $\in [0.1, 0.5]$ with a stride of 0.1. Under the optimal configuration

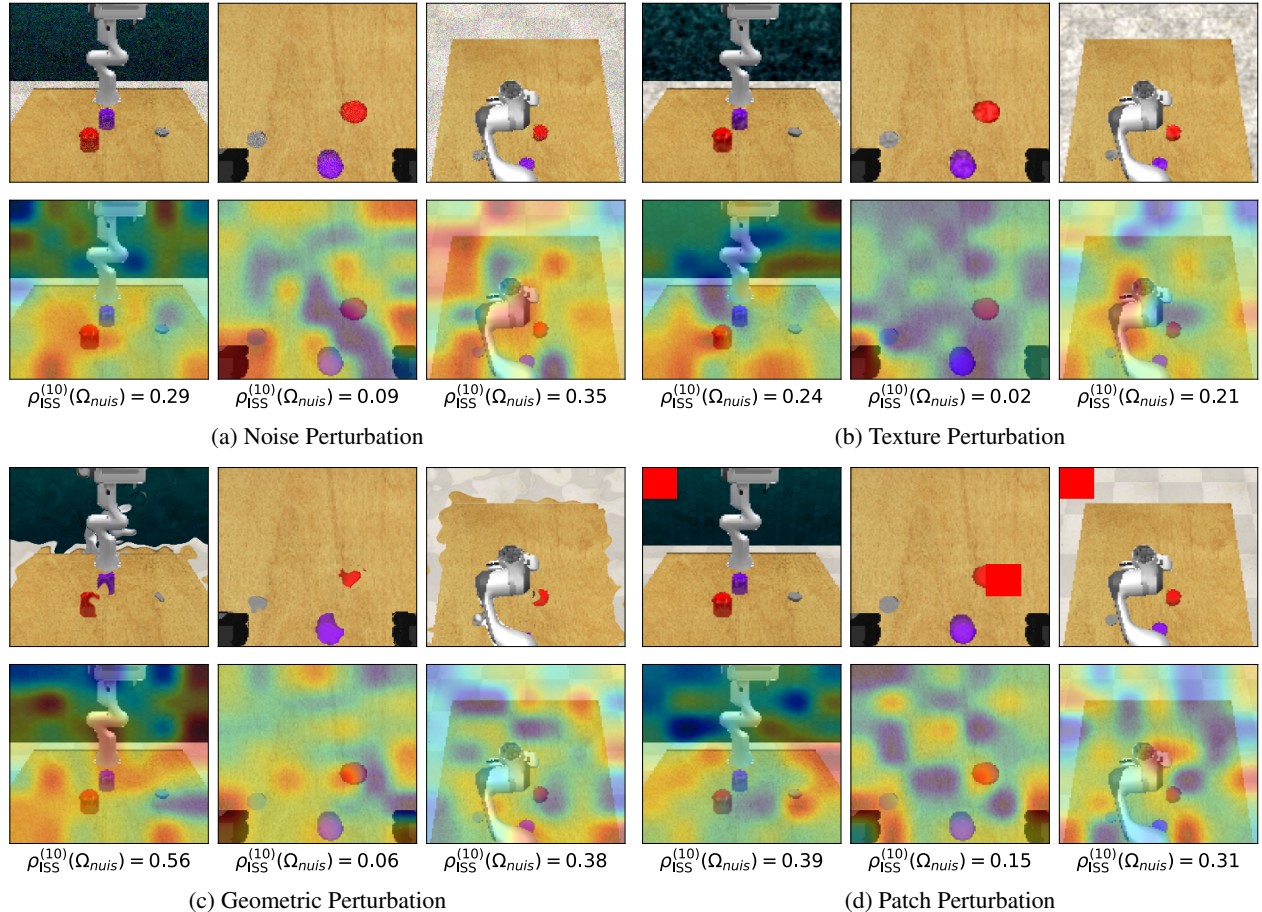

$\rho_{\text{ISS}}^{(10)}(\Omega_{nuis}) = 0.29$    $\rho_{\text{ISS}}^{(10)}(\Omega_{nuis}) = 0.09$    $\rho_{\text{ISS}}^{(10)}(\Omega_{nuis}) = 0.35$      $\rho_{\text{ISS}}^{(10)}(\Omega_{nuis}) = 0.24$    $\rho_{\text{ISS}}^{(10)}(\Omega_{nuis}) = 0.02$    $\rho_{\text{ISS}}^{(10)}(\Omega_{nuis}) = 0.21$

(a) Noise Perturbation            (b) Texture Perturbation

$\rho_{\text{ISS}}^{(10)}(\Omega_{nuis}) = 0.56$    $\rho_{\text{ISS}}^{(10)}(\Omega_{nuis}) = 0.06$    $\rho_{\text{ISS}}^{(10)}(\Omega_{nuis}) = 0.38$      $\rho_{\text{ISS}}^{(10)}(\Omega_{nuis}) = 0.39$    $\rho_{\text{ISS}}^{(10)}(\Omega_{nuis}) = 0.15$    $\rho_{\text{ISS}}^{(10)}(\Omega_{nuis}) = 0.31$

(c) Geometric Perturbation           (d) Patch Perturbation

*Figure 6.* **ISS Visualization under perturbations.** Saliency maps for the *Close Jar* task are visualized under four interventions (Gaussian noise, texture, geometric, and patch) at a fixed timestep. We report the variations of the nuisance mass ratio (nmr@10) under different perturbations.

$(N = 100, p = 0.3)$, the model records a total latency of 5.18s and a throughput of 0.19Hz (Tab. 2, Appendix B).

Although more expensive than single-pass heuristics such as attention weights, the proposed Monte Carlo strategy is substantially more efficient than exhaustive causal intervention, making causal analysis feasible for high-dimensional VLAs. Concretely, for 3 views with $16 \times 16$ tokens over $T$ steps, exhaustive intervention requires $\mathcal{O}(T \times 3 \times 16^2)$ forward passes, whereas our method reduces the cost to $\mathcal{O}(T \times N)$. Consequently, while the current throughput, for example 0.39Hz at $N = 50$, is insufficient for high-frequency real-time control, it remains well-suited for offline safety verification and post hoc interpretability.

**Visualization.** We present saliency maps (heatmaps) under four perturbation paradigms, including noise, texture, geometry, and patch, evaluated using the nuisance mass ratio at Top-10% (nmr@10). The detailed implementations of these perturbations are provided in Appendix F.

Figure 6 demonstrates that ISS concentrates saliency on manipulation-relevant components in the wrist view, maintaining low nuisance attribution with $\rho_{nuis}$ values of $0.09, 0.02, 0.06, 0.15$ under noise, texture, geometric, and patch perturbations, respectively. In contrast, the front view shifts focus toward robot links or background ($\rho_{nuis} \in \{0.29, 0.24, 0.56, 0.39\}$), while the overhead view exhibits diffuse activations across the workspace ($\rho_{nuis} \in \{0.35, 0.21, 0.38, 0.31\}$).

## 6. Limitations and Future Work

**Limitations.** First, the current ISS formulation implicitly relies on a unimodal decision assumption, where action prediction error serves as a faithful surrogate for distributional divergence. In environments with multimodal action distributions or ambiguous long-horizon planning, different modes may respond heterogeneously to visual interventions, causing mean-based or point-estimate errors to collapse mode-specific causal effects, thereby invalidating the ISS attribution. Second, although Monte Carlo sampling sub-

stantially reduces the combinatorial cost compared to exhaustive enumeration, ISS still requires multiple forward or backward passes per state. This computational overhead is significantly higher than that of attention-based saliency methods, which yield explanations in a single inference, preventing ISS from being used for online diagnosis and limiting it to post hoc, offline analysis. Third, the accuracy of the NMR metric depends critically on the quality of nuisance masks. While simulation environments provide precise ground-truth masks, real-world deployment typically relies on instance segmentation models, which increases data-collection costs and makes segmentation errors and prompt design more sensitive, directly propagating into NMR estimation bias.

**Future Work.** First, we aim to improve the computational efficiency of ISS by replacing Monte Carlo aggregation with low-variance estimators and local approximations, such as Hutchinson-style stochastic trace estimation to compress global token attribution into one to two backward passes, first-order interventional approximations that recast ISS as a single-pass Jacobian-based score, or hybrid schemes that move Monte Carlo sampling entirely offline while enabling deterministic, single-inference scoring online via learned gating mechanisms. Second, we plan to relax the unimodality assumption by extending ISS to multimodal settings, for example, by defining mode-aware or mixture-consistent ISS variants and by lifting error proxies from raw action space to latent action representations where multimodal structure is more explicitly encoded. Third, to reduce dependence on explicit masks, we will explore self-supervised or weakly supervised nuisance discovery that infers causal partitions directly from intervention responses, rather than relying on external segmentation models. Finally, we will conduct large-scale evaluations across diverse VLA architectures and benchmarks to assess the robustness, scalability, and generality of ISS and NMR beyond a single model family.

## 7. Conclusion

In this paper, we introduced the Interventional Significance Score (ISS) to estimate the causal influence of visual regions on action predictions and the Nuisance Mass Ratio (NMR) to quantify reliance on task-irrelevant features. Empirically, NMR tracks policy generalization under distribution shifts, as indicated by its strong negative correlation with task success rate on AGNOSTOS. Together, these results establish interventional attribution as a principled diagnostic framework for evaluating causal alignment in embodied policies.

## Impact Statement

This paper introduces a diagnostic framework (ISS and NMR) for measuring the causal influence of visual features on decisions made by Vision–Language–Action (VLA) policies. By enabling the identification of reliance on spurious visual correlations and causal misalignment, our approach may support the development of more reliable and robust embodied systems, including robotics and autonomous agents, particularly in settings where safety and generalization under distribution shifts are critical.

At the same time, interpretability and causal diagnostics carry risks if they are over-trusted or misinterpreted as formal guarantees; attribution scores are approximate and do not certify the correctness of a model's reasoning. There is also a risk of misuse if attribution tools are applied in high-stakes domains without appropriate human oversight, or used to justify opaque decision-making rather than improve it. Additionally, repeated interventional evaluations incur computational overhead, which may limit applicability in resource-constrained environments.

We therefore view ISS and NMR as diagnostic tools that should be used alongside complementary evaluation methods, uncertainty analyses, and human judgment, with careful consideration of the contexts in which they are applied.

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

# A. Theoretical Derivations and Validity Constraints

## A.1. Stochastic Estimation of Interventional Significance

**Problem Formulation:** Quantifying the interventional importance of a specific visual token $i$ theoretically requires evaluating the loss function $\mathcal{L}$ across the combinatorial space of all possible context configurations. Let $\mathbb{S} = \{0, 1\}^M$ denote the space of all possible binary masks of $\mathbf{X}$, where $M = n_1 + n_2$ is defined in Eq. (2). We model the "coalitional context" as a random vector $\mathbf{m} \in \mathbb{S}$ governed by the product measure $\boldsymbol{\mu}_p$, where each component $m_j \sim \text{Bernoulli}(p)$ represents the retention probability. Crucially, the loss term $\mathcal{L}(\mathbf{X}, \mathbf{m})$ measures the deviation of the policy's predicted action from the ground truth action $\mathbf{a}^*$, evaluated under partial input occlusion using teacher forcing (Williams & Zipser, 1989). We treat $\mathcal{L}(\mathbf{X}, \mathbf{m})$ as a deterministic functional of the masked input, with all stochasticity arising solely from the random mask sampling process.

**Theoretical Definition (Global Occlusion Sensitivity):** We first define $\Psi_i(p)$, as the expected failure of the policy given that token $i$ is forcibly occluded ($m_i = 0$). This metric marginalizes over the combinatorial masking configurations of all other tokens:

$$\Psi_i(p) = \mathbb{E}_{\mathbf{m} \sim \boldsymbol{\mu}_p} \left[ \mathcal{L}(\mathbf{X}, \mathbf{m}) \mid m_i = 0 \right]. \tag{11}$$

This definition captures the "absolute necessity" of token $i$: a high $\Psi_i$ indicates that the policy consistently deviates from the ground truth whenever $i$ is **absent**, regardless of the remaining context $\mathbf{m}_{\backslash i}$. Unlike contrastive causal effects, this absolute measure is robust to baseline performance shifts when used strictly for *ranking* critical features. For a fixed task and masking distribution $\boldsymbol{\mu}_p$, ranking tokens by $\Psi_i(p)$ is equivalent to ranking by any affine-transformed contrastive score (e.g., $\Psi_i(p) - \mathbb{E}[\mathcal{L}]$), and therefore does not require explicit centering. Consequently, the Interventional Significance Score (ISS) for token $i$ is derived as the rise in risk relative to the baseline (or the limit as $p \to 1$), isolating the marginal contribution of the token.

**Derivation of the Unbiased Estimator:** Direct computation of Eq. (11) is intractable. We derive the estimator used in Algorithm 1 via Monte Carlo integration. Let $\{\mathbf{m}^{(k)}\}_{k=1}^N$ be $N$ i.i.d. samples drawn from $\boldsymbol{\mu}_p$.

A standard Monte Carlo approach might employ a Ratio Estimator, dividing the sum of losses by the empirical count of masks where $m_i = 0$. However, Ratio Estimators are known to exhibit finite-sample bias and increased variance in practice, and are prone to numerical instability when the empirical count of $m_i = 0$ is low or zero.

To ensure statistical rigor, we implement a fixed-denominator estimator. We define the estimator using the known theoretical probability mass of the occlusion event, $P(m_i = 0) = 1 - p$:

$$\widehat{\text{ISS}}_i(N, p) = \frac{1}{N(1-p)} \sum_{k=1}^N (1 - m_i^{(k)}) \cdot \mathcal{L}(\mathbf{X}, \mathbf{m}^{(k)}) \tag{12}$$

where $(1 - m_i^{(k)})$ acts as the indicator function $\mathbb{I}[m_i^{(k)} = 0]$. This formulation explicitly avoids the numerical singularity associated with small denominators while ensuring unbiasedness.

**Proof of Unbiasedness:** We prove that Eq. (12) is an unbiased estimator of the theoretical risk $\Psi_i(p)$. Taking the expectation of the estimator:

$$\begin{aligned}
\mathbb{E}[\widehat{\text{ISS}}_i] &= \frac{1}{N(1-p)} \sum_{k=1}^N \mathbb{E}\left[ (1 - m_i^{(k)}) \mathcal{L}(\mathbf{X}, \mathbf{m}^{(k)}) \right] \\
&= \frac{1}{N(1-p)} \cdot N \cdot P(m_i = 0) \cdot \mathbb{E}[\mathcal{L} \mid m_i = 0] \\
&= \frac{N(1-p)}{N(1-p)} \Psi_i(p) = \Psi_i(p)
\end{aligned} \tag{13}$$

Thus, our method provides a mathematically rigorous estimate of the global sensitivity without the finite-sample bias inherent in Ratio Estimators.

**Probabilistic Interpretation of the Measure Space:** It is critical to note that Eq. (12) approximates the expectation over the *weighted* subspace defined by $\boldsymbol{\mu}_p$, rather than a uniform summation over the power set $2^M$. This focus effectively prioritizes likely coalitional structures governed by the sparsity parameter $p$. Using the fixed probability mass $(1 - p)$ in the denominator, the estimator correctly scales the accumulated contributions to the density of the sampled subspace.

## A.2. Action MSE as a Proxy for KL Divergence

**Continuous Policy Formulation.** We model the VLA policy $\pi_\theta(\mathbf{a}|\mathbf{X})$ as a continuous probability density function over the kinematic control space $\mathcal{A}$. Specifically, we adopt the standard assumption in regression-based imitation learning that the policy output follows a multivariate Gaussian distribution with a parameterized mean $\boldsymbol{\mu}_\theta(\mathbf{X})$ and a fixed, isotropic covariance matrix $\Sigma = \sigma^2 \mathbf{I}$:

$$\pi_\theta(\mathbf{a}|\mathbf{X}) = \mathcal{N}(\mathbf{a}; \boldsymbol{\mu}_\theta(\mathbf{X}), \sigma^2 \mathbf{I}) \tag{14}$$

This formulation satisfies the prerequisite for applying the Fisher Information metric in unbounded continuous spaces.

**Derivation of the MSE-KL Equivalence.** We aim to quantify the divergence between the original policy distribution $\pi(\mathbf{a}|\mathbf{X})$ and the perturbed policy distribution $\pi(\mathbf{a}|\mathbf{X} + \delta)$, where $\delta$ represents a nuisance perturbation injected into the observation space $\mathcal{O}$. The Kullback-Leibler (KL) divergence between two multivariate Gaussian distributions $\mathcal{P} = \mathcal{N}(\boldsymbol{\mu}_1, \Sigma)$ and $\mathcal{Q} = \mathcal{N}(\boldsymbol{\mu}_2, \Sigma)$ is analytically defined as:

$$D_{\mathrm{KL}}(\mathcal{P}\|\mathcal{Q}) = \frac{1}{2}\left[(\boldsymbol{\mu}_1 - \boldsymbol{\mu}_2)^\top \Sigma^{-1}(\boldsymbol{\mu}_1 - \boldsymbol{\mu}_2) + \mathrm{tr}(\Sigma^{-1}\Sigma) - d + \ln\frac{|\Sigma|}{|\Sigma|}\right] \tag{15}$$

Under the assumption of homoscedasticity utilized in our setup, the trace ($\mathrm{tr}(\Sigma^{-1}\Sigma) = d$) and logarithmic terms cancel out. Substituting $\Sigma = \sigma^2 \mathbf{I}$ simplifies the equation to:

$$D_{\mathrm{KL}}(\pi(\mathbf{a}|\mathbf{X})\|\pi(\mathbf{a}|\mathbf{X} + \delta)) = \frac{1}{2\sigma^2}\|\boldsymbol{\mu}_\theta(\mathbf{X}) - \boldsymbol{\mu}_\theta(\mathbf{X} + \delta)\|_2^2 \propto \mathrm{MSE} \tag{16}$$

Under the assumed density regime, this derivation demonstrates that minimizing the mean squared error (MSE) between the predicted action means is theoretically equivalent to minimizing the KL divergence in the probabilistic log-likelihood space.

**Geometric Interpretation via Fisher Information.** From the perspective of Information Geometry (Amari, 2016), the Fisher Information Matrix (FIM) $\mathcal{I}(\theta)$ defines the local Riemannian metric of the statistical manifold. For the defined Gaussian family, the FIM with respect to the mean parameters corresponds to the inverse covariance matrix $\mathcal{I} = \Sigma^{-1}$. Since the action space $\mathcal{A}$ undergoes statistical whitening, the control dimensions are effectively decorrelated, justifying the local Euclidean approximation $\mathcal{I} \approx \mathbf{I}$. Consequently, the Euclidean distance (MSE) in the action space serves as a faithful **first-order consistency proxy** for the distributional shift.

**Regime of Validity and Limitations.** We strictly define the validity of this proxy under the **Unimodal Trajectory Assumption**. A VLA task may admit multiple episodes with varying initial object positions and different trajectories, while the policy remains concentrated around a single dominant action trend without branching behaviors, as shown in Figure 7. The equivalence $D_{\mathrm{KL}} \propto \mathrm{MSE}$ holds when the policy approximates a single mode of behavior. We acknowledge that in scenarios involving high ambiguity or multi-modal global planning, a Gaussian approximation may average conflicting modes, rendering MSE an insufficient metric. However, for the specific purpose of assessing robustness against non-causal visual nuisances in deterministic execution phases, this metric remains both mathematically sound and computationally efficient.

# B. Computation Efficiency Results

We demonstrate the full computational efficiency results of ISS, as shown in Tab. 2. The computational cost scales linearly with the sample size $N$, where the latency increases from 2.59s (0.39Hz) at $N = 50$ to 10.35s (0.10Hz) at $N = 200$, corresponding to a slowdown factor rising from $33.0\times$ to $131.7\times$. In contrast, these metrics remain invariant to the mask ratio $p$ across the interval $[0.1, 0.5]$.

# C. Training Details and Results (VLA model $\pi_{0.5}$ )

## C.1. Training Description

**Training Data.** We use the AGNOSTOS dataset for training and evaluation, comprising 41 manipulation tasks. The training split consists of 18 seen tasks, each with 200 episodes, for a total of 3,600 episodes (approximately 140 GB) distributed across five files. Evaluation is conducted on 23 unseen tasks, with 25 episodes per task, totaling 575 episodes (20.2 GB) provided in a single file for cross-task generalization testing. Each episode is annotated with four natural-language prompts

*Table 2.* Inference latency (s), throughput (Hz), and slowdown factor ($\times$) relative to single inference (0.079 s / 12.7 Hz). The results demonstrate that computational cost scales linearly with $N$ while remaining invariant to the mask ratio $p$.

| **Ratio** ($p$) | N = 50 | | | N = 100 | | | N = 150 | | | N = 200 | | |
|---|---|---|---|---|---|---|---|---|---|---|---|---|
| | Lat (s) | Hz | Slow | Lat (s) | Hz | Slow | Lat (s) | Hz | Slow | Lat (s) | Hz | Slow |
| 0.1 | 2.59 | 0.39 | 33.0 | 5.04 | 0.20 | 64.1 | 7.67 | 0.13 | 97.5 | 10.34 | 0.10 | 131.5 |
| 0.2 | 2.59 | 0.39 | 33.0 | 5.18 | 0.19 | 65.8 | 7.77 | 0.13 | 98.8 | 10.39 | 0.10 | 132.1 |
| 0.3 | 2.59 | 0.39 | 33.0 | 5.18 | 0.19 | 65.8 | 7.77 | 0.13 | 98.8 | 10.36 | 0.10 | 131.8 |
| 0.4 | 2.61 | 0.38 | 33.2 | 5.19 | 0.19 | 66.0 | 7.76 | 0.13 | 98.8 | 10.35 | 0.10 | 131.7 |
| 0.5 | 2.59 | 0.39 | 33.0 | 5.20 | 0.19 | 66.2 | 7.76 | 0.13 | 98.7 | 10.35 | 0.10 | 131.7 |

for VLA training. All data are generated in RLBench (James et al., 2020) with a simulation timestep of 50 ms. For VLA model learning, the dataset is converted to LeRobot format (Cadene et al., 2024) to ensure compatibility with the training pipeline, using RGB observations from the `front_rgb`, `overhead_rgb`, and `wrist_rgb` streams. All episodes are replayed in RLBench to obtain frame-level segmentation masks, as illustrated in Figure 8. For each frame, the simulator directly segments three regions, corresponding to $\rho_{act}$, $\rho_{sup}$, and $\rho_{nuis}$, and saves the resulting masks as `front_mask`, `overhead_mask`, and `wrist_mask`.

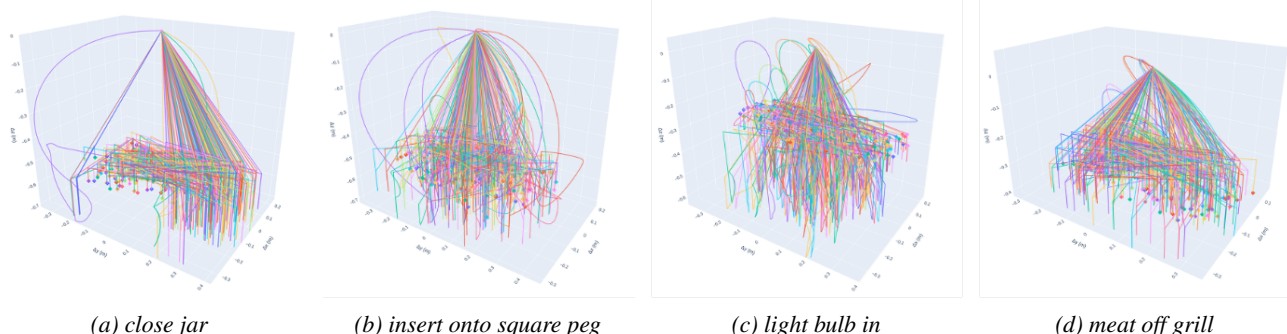

| *(a) close jar* | *(b) insert onto square peg* | *(c) light bulb in* | *(d) meat off grill* |

*Figure 7.* **Visualization of Episode Trajectories.** Each plot represents the aggregated 3D spatial paths of the end-effector recorded across all episodes. The subfigures (a) through (d) illustrate four distinct manipulation tasks in the AGNOSTOS dataset.

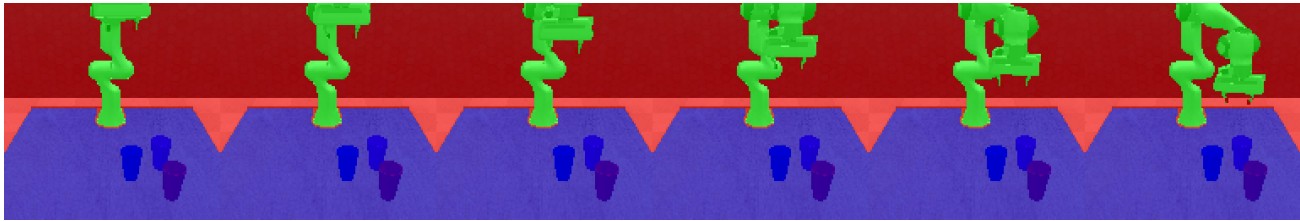

*Figure 8.* Segmentation masks for the *"close jar"* task (episode 0), showing 6 frames from the front view. Green indicates action-relevant regions ($\Omega_{act}$), blue indicates task-support regions ($\Omega_{sup}$), and red indicates nuisance regions ($\Omega_{nuis}$).

**Fine-tuning.** Implemented in the JAX framework, we fine-tuned the models using supervised fine-tuning (SFT) for 20 epochs on a single 96 GB NVIDIA RTX 6000 GPU with a batch size of 64. We adopted Low-Rank Adaptation (LoRA) targeting both attention and feed-forward network (FFN) layers, utilizing distinct rank configurations adapted to the model scale: $r = 16$ for the vision-language model expert (Gemma 2B) and $r = 32$ for the action expert (Gemma 300M). We explicitly capped GPU memory usage at approximately 90% of the available VRAM (about 90 GB), under which the full training process took around 2 hours, corresponding to a total of 1,000 optimization steps. All experiments are conducted using `bfloat16` precision.

The model is initialized from a pre-trained checkpoint *pi05_base* with the *Franka Emika Panda robotic arm* asset and three camera views (front, overhead, and wrist). The VLA $\pi_{0.5}$ model takes the RGB observations $x \in \mathbb{R}^{H \times W \times 3}$ from three camera views (front, overhead, and wrist) as input, with all images and prompts sourced from the AGNOSTOS

dataset (Zhou et al., 2025a). Figure 9 reports the training dynamics of $\pi_{0.5}$ over 1,000 optimization steps. The training loss decreases sharply from approximately 1.6 to below 0.2 within the first 100 steps and continues to gradually converge, reaching around 0.1 by the end of fine-tuning. Meanwhile, the parameter $\ell_2$ norm increases smoothly from approximately 1803.87 to 1803.92 over the course of training, without abrupt spikes or oscillations, indicating controlled parameter updates throughout optimization.

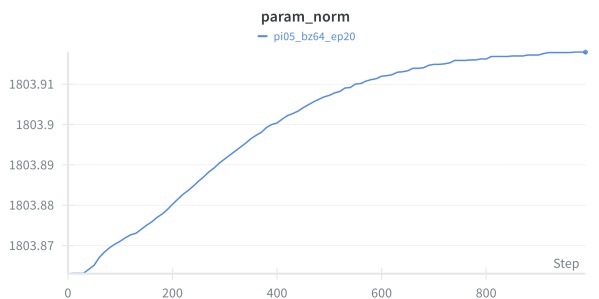

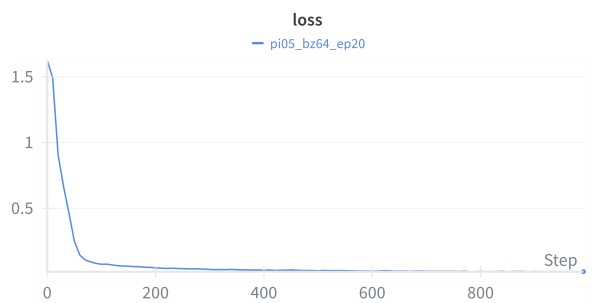

*(a)* **Parameter norm.** X-axis: training steps. Y-axis: $\ell_2$ norm of model parameters.

*(b)* **Training loss.** X-axis: training steps. Y-axis: objective loss value.

*Figure 9.* Optimization dynamics during VLA $\pi_{0.5}$ training.

### C.2. Implementation and Evaluation.

**Implementation.** We represent actions as continuous 8-DoF control signals. VLA $\pi_{0.5}$ directly models continuous-valued actions via a flow-matching formulation, without discretizing the action space or requiring post-hoc decoding. The control frequency is set to 20 Hz (0.05 s per step), while policy inference is performed every 8 control steps, corresponding to an inference frequency of 2.5 Hz. At each inference step, the model predicts a fixed-length chunk of 16 future 8-DoF actions, where the first seven dimensions correspond to continuous joint angles and the last dimension encodes the gripper open/close command. During execution, we run the environment at 20 Hz and refresh observations at every control step. Policy inference is triggered every 8 control steps; when invoked, the model predicts a 16-step action chunk, from which we execute only the first 8 actions sequentially over the next 8 control steps before the next inference. After completing an episode, the resulting per-step rollouts are rendered into a video.

**Online Evaluation.** We evaluate $\pi_{0.5}$ across 5 distinct random seeds (0, 13, 50, 77, 99) covering 41 diverse manipulation tasks. Each task is tested with 25 independent trials in newly generated scenes, whose object spatial configurations differ from those in the fixed training dataset. Quantitative results in Tab. 3, determined by strict sensor-based success conditions (e.g., proximity checks in `DetectedCondition`), show that the policy achieves a mean success rate of $59.07 \pm 0.58\%$ on seen tasks, with 100% success on primitives such as *"meat off grill"* across all initialization seeds. For zero-shot generalization to unseen tasks absent from the training set, the agent attains an average success rate of $24.00 \pm 1.92\%$, specifically achieving $28.74 \pm 1.97\%$ on Level 1 tasks with similar semantics and $16.96 \pm 1.21\%$ on structurally novel Level 2 tasks. Figure 13 shows representative successful and failed episodes on seen and unseen tasks.

## D. Detailed Results for ISS and NMR Evaluation

### D.1. Quantitative Results

**Top-$k$ Sensitivity Results.** For top-$k$ computation, we flatten the 2D ISS heatmap to extract indices of the highest-scoring pixels and compute the intersection ratio via Equation 10. Figure 10 illustrates the correlation analysis across sensitivity thresholds $k \in \{1, 5, 10, 15, 20\}$. The results reveal a robust negative correlation, where higher causal attribution to nuisance features predicts lower task success rates. Since the strongest signal occurs at $k = 10$ with $r = -0.77$, we adopt nmr@10 as the default metric for subsequent experiments, noting that extreme sparsity at $k = 1$ or excessive dilution at $k = 20$ slightly attenuates this correlation.

**Fidelity Results.** Figure 11 illustrates the fidelity analysis stratified by *Seen* and *Unseen* task splits, presenting detailed scatter plots that visualize the correlation between saliency changes and action deviations across three hard (structured) interventions. We benchmark ISS against saliency maps derived from attention scores (ATT) and token norms (NORM)

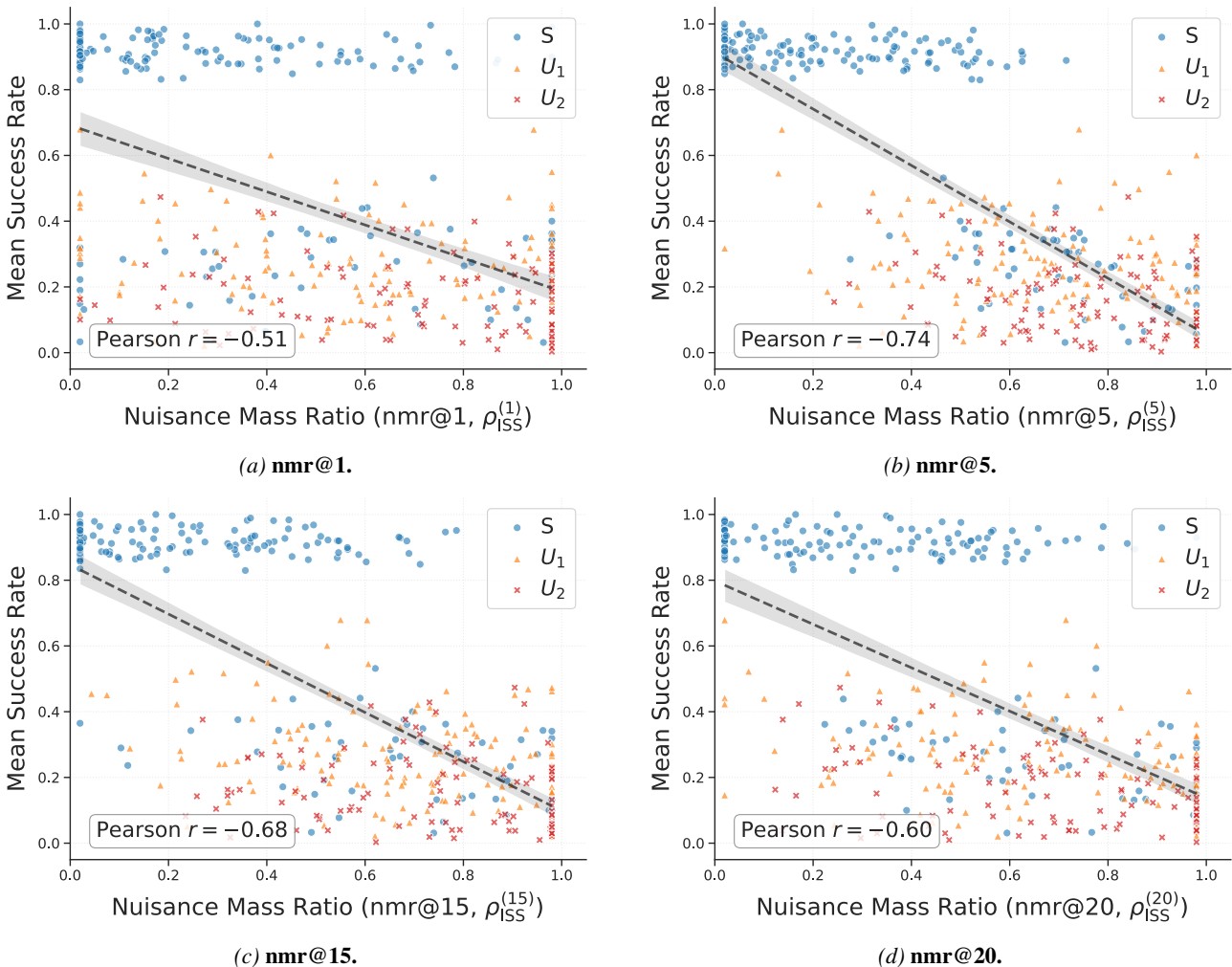

*Figure 10.* Correlation between nuisance mass ratio (nmr@k) and task success rate under different cutoff values $k$. The case $k{=}10$ is omitted here as it yields the strongest correlation and is presented in the main paper.

under identical metric protocols. Scatter plots confirm that ISS maintains superior linearity across all splits. Under the *Unseen* geometric (GEO) intervention, ISS achieves a Pearson coefficient of $r = 0.79$, significantly outperforming ATT ($r = 0.60$) and NORM ($r = 0.48$). This provides empirical evidence that our proposed ISS faithfully captures the policy's reliance on geometric cues, thereby ensuring superior fidelity compared to magnitude-based baselines.

### D.2. Qualitative Results

Figure 12 visualizes heatmaps for the "close jar" task across multiple views to compare token norm $\phi = |\cdot|$, attention score $\phi = $ ATT, and Interventional Significance Score $\phi = $ ISS. While NORM and ATT baselines exhibit high dispersion across task-irrelevant background textures, ISS demonstrates superior localization of causal entities. Specifically, the front view ISS highlights the robot manipulator, whereas the wrist view focuses on the active gripper fingers and the target jar lid. Within the $\pi_{0.5}$ VLM expert, $\rho_{|\cdot|}^{(10)}(\Omega_{\text{nuis}})$ and $\rho_{\text{ATT}}^{(10)}(\Omega_{\text{nuis}})$ show severe spatial dispersion over nuisance regions (e.g., $\rho_{|\cdot|}^{(10)} = 0.76$ in the front view). This discrepancy reveals a misalignment where computational allocation defined by $\phi = |\cdot|$ and $\phi = $ ATT diverges from the actual causal features identified by $\phi = $ ISS. Consequently, although $\pi_{0.5}$ effectively learns the task-specific causal structure, it retains a residual dependence on irrelevant environmental contexts.

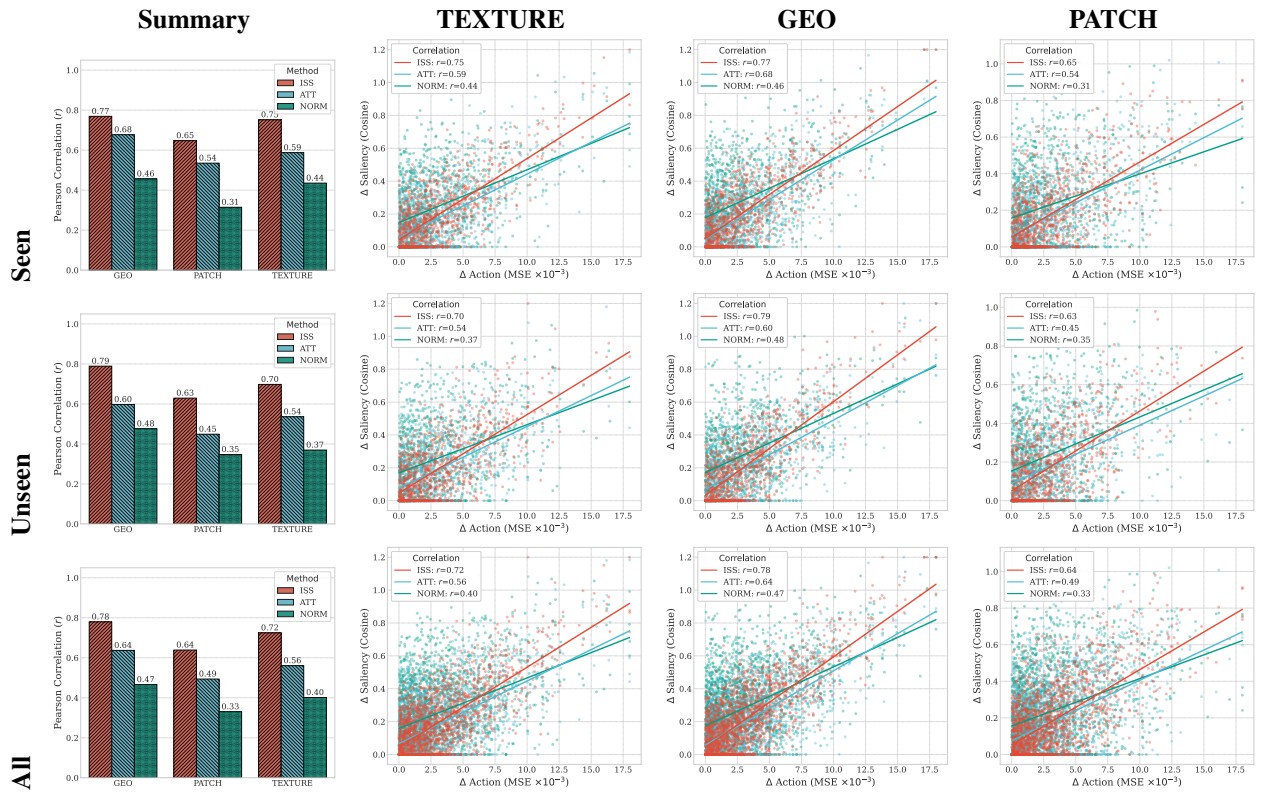

*Figure 11.* **Saliency Fidelity Analysis** between Δ action and Δ saliency across evaluation splits and perturbation types. Rows correspond to evaluation splits (Seen, Unseen, All), while columns correspond to different analysis settings. The first column summarizes Pearson correlation coefficients across perturbations. The remaining columns present scatter plots for ISS, ATT, and NORM under TEXTURE, GEO, and PATCH perturbations, together with linear fits.

## E. Baseline Extraction and Metric Calculation Details

### E.1. Architecture

The VLA $\pi_{0.5}$ architecture integrates a Vision Encoder, a Gemma Language Model, and a specialized Action Expert. Specifically, it employs **SigLIP** (Zhai et al., 2023) as the visual encoder to process RGB observations into visual tokens, and **PaliGemma** (Beyer et al., 2024) as the multimodal language-model backbone that integrates these visual tokens with textual instructions. To evaluate the intrinsic visual grounding capabilities of the baselines, we extract features and attention weights directly from the SigLIP encoder layers, prior to the cross-modal fusion in the PaliGemma decoder.

### E.2. Attention Score (ATT)

The Attention Score baseline quantifies the significance of each visual patch using the self-attention mechanism in the Vision Encoder.

**Extraction Process:** We extract the attention weights $A \in \mathbb{R}^{H \times N \times N}$ from the final transformer block (Block 26) of the SigLIP encoder. We select this specific layer to capture the most abstract semantic features aligned with the language instruction, effectively representing the highest-level visual processing before VLA integration. Here, $H$ denotes the number of attention heads, and $N$ represents the total number of tokens. The raw attention matrix is computed as:

$$A = \text{softmax}\left(\frac{QK^T}{\sqrt{d_k}}\right)$$

where $Q, K \in \mathbb{R}^{N \times d_k}$ are the query and key projections of the visual tokens.

**Aggregation and Scoring:** To derive a scalar importance score $S_{\text{ATT}}^{(i)}$ for the $i$-th visual token, we compute the mean across

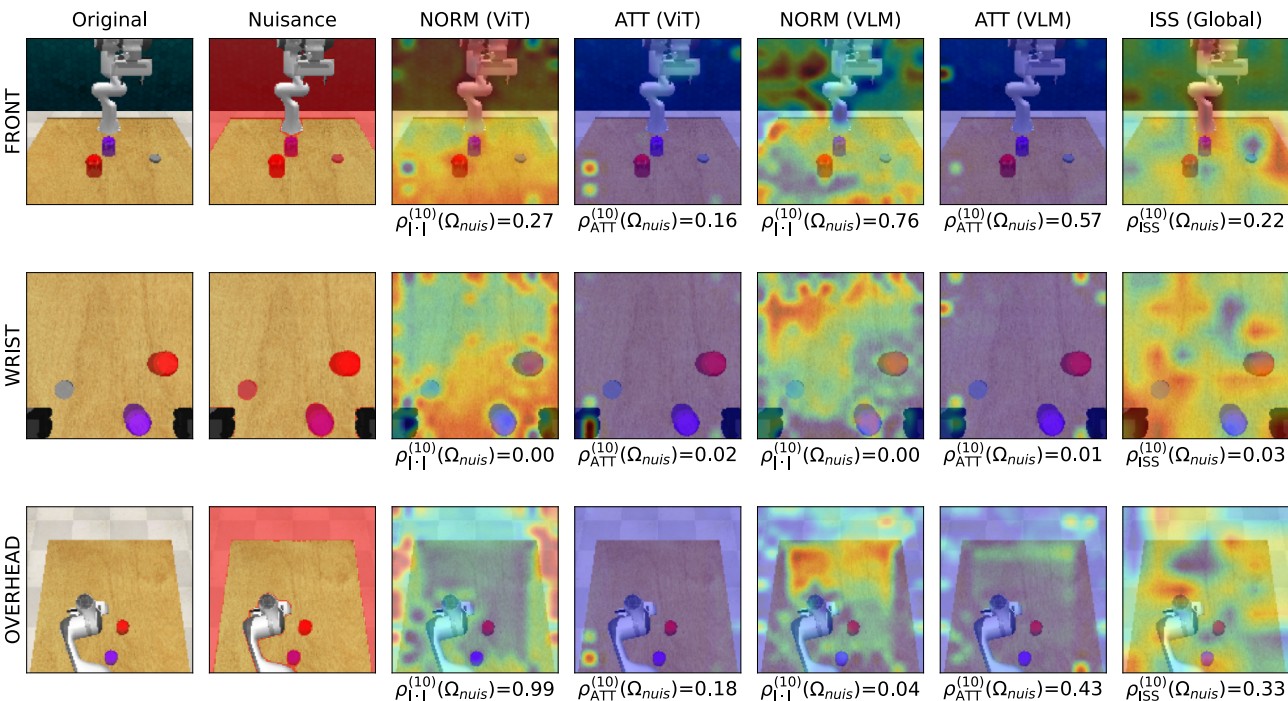

*Figure 12.* Visualization of saliency maps and nuisance mass ratio $\rho_\phi^{(k)}(\Omega_{nuis})$ with $k = 10\%$ under token magnitude ($\phi = |\cdot|$), attention score ($\phi = $ ATT), and interventional significance score ($\phi = $ ISS) for the "close jar" task across different camera views (Front, Wrist, Overhead).

all attention heads to generate a unified attention map $\bar{A} \in \mathbb{R}^{N \times N}$. Since the visual tokens in the backbone of VLA $\pi_{0.5}$ utilize prefix attention mechanisms, allowing bidirectional communication within the image sequence, we quantify the total attention mass received by token $i$ by summing the weights along the query dimension:

$$S_{\text{ATT}}^{(i)} = \sum_{j=1}^{N} \bar{A}_{j,i}$$

This calculation treats the attention matrix as an adjacency matrix of a directed graph, where the score reflects the centrality or popularity of a visual patch within the backbone's internal representation.

### E.3. Token Norm (NORM)

The Token Norm baseline assumes that the magnitude of the feature vector correlates with the informational content or activation intensity of the token.

**Extraction Process:** We extract the final output embeddings $Z \in \mathbb{R}^{N \times D}$ from the SigLIP encoder, where $D$ is the embedding dimension.

**Scoring:** The importance score $S_{\text{NORM}}^{(i)}$ for the $i$-th token is calculated as the $L_2$ norm of its embedding vector $z_i$:

$$S_{\text{NORM}}^{(i)} = \|z_i\|_2 = \sqrt{\sum_{d=1}^{D} (z_{i,d})^2}$$

Tokens with higher norms are hypothesized to have a stronger influence on subsequent layers of the VLA network, given the nature of Layer Normalization and residual connections.

## E.4. Saliency Map Generation and Metric Calculation

**Heatmap Generation:** To generate the heatmap, we first isolate the spatial visual tokens from the derived scores $S$. We explicitly exclude non-spatial tokens (e.g., registers or special start/end tokens) to ensure the remaining $N_{spatial}$ tokens correspond strictly to the image grid. The filtered scores are reshaped into a 2D grid of size $h \times w$ (where $h \times w = N_{\text{spatial}}$). This grid is then upsampled to the original image resolution $H_{img} \times W_{img}$ via bilinear interpolation to produce the continuous saliency map $M$.

**Nuisance Sensitivity Metric ($\rho$):** To quantify the susceptibility of the model to task-irrelevant features, we calculate the Nuisance Sensitivity metric $\rho_{nuis}$. Given a binary ground-truth mask $B_{nuis}$ where 1 indicates a nuisance region (e.g., a distractor object or background noise) and 0 indicates task-relevant areas, the metric is defined as the proportion of total saliency mass falling within the nuisance region:

$$\rho_{nuis} = \frac{\sum_{x,y} M_{x,y} \cdot B_{nuis,x,y}}{\sum_{x,y} M_{x,y}}$$

A higher $\rho_{nuis}$ value indicates that the baseline method (ATT or NORM) incorrectly assigns higher importance to nuisance features, thereby failing to distinguish between causal and non-causal visual signals.

# F. Intervention and Implementation

## F.1. ISS Computation and Soft Interventions

**Bernoulli Mask.** To compute the **Interventional Significance Score (ISS)** for VLA $\pi_{0.5}$, we employ a randomized masking procedure rooted in the RISE algorithm (Petsiuk et al., 2018). We generate a set of binary masks $\mathbf{m} \in \{0,1\}^M$ using hyperparameters optimized for convergence ($N = 100, p = 0.3$). These masks are sampled on a coarse $7 \times 7$ grid and subsequently upsampled to the target resolution to ensure the perturbations maintain spatial coherence rather than pixel-level noise. For each time step $t$, the perturbed observation $\tilde{\mathbf{V}}_{t,k}$ fuses the original image $\mathbf{V}_t$ and a Gaussian-blurred version $\mathbf{V}_t^{blur}$ using the mask $\mathbf{m}_k$:

$$\tilde{\mathbf{V}}_{t,k} = \mathbf{V}_t \odot \mathbf{m}_k + \mathbf{V}_t^{blur} \odot (\mathbf{1} - \mathbf{m}_k) \tag{17}$$

We quantify the significance of the occluded regions by accumulating the $L_2$ divergence between the baseline action $\mathbf{a}_t^*$ and the perturbed action predicted by $\pi_{0.5}$. The final saliency map $\mathbf{S}_t$ is the normalized average of these weighted divergences:

$$\mathbf{S}_t = \frac{1}{N(1-p)} \sum_{k=1}^{N} \|\pi_{0.5}(\tilde{\mathbf{V}}_{t,k}) - \mathbf{a}_t^*\|_2^2 \cdot (\mathbf{1} - \mathbf{m}_k) \tag{18}$$

**Gaussian Noise.** To assess the robustness of $\pi_{0.5}$ against irrelevant visual changes, we introduce additive Gaussian noise only in the nuisance regions identified as **Soft Interventions**. We sample noise $\epsilon \sim \mathcal{N}(0, 0.25)$ and inject it into the normalized observation $\mathbf{V} \in [0,1]$ exclusively where the semantic mask indicates non-essential data (denoted as $\mathbf{m}_{sem} = 0$). Notably, this noise configuration employs the same standard deviation ($\sigma = 0.25$) as the Gaussian perturbation used in our ISS computation to generate the perturbed (blurred) observations. While ISS applies this perturbation via randomized masking to probe feature significance, we restrict noise injection here to nuisance regions to strictly evaluate policy resilience against background distribution shifts of equal magnitude. The noisy observation $\mathbf{V}_{noisy}$ is computed via pixel-wise clipping to maintain valid intensity ranges:

$$\mathbf{V}_{noisy} = \text{clip}(\mathbf{V} + \epsilon, 0, 1) \odot \mathbb{I}_{\{\mathbf{m}_{sem}=0\}} + \mathbf{V} \odot \mathbb{I}_{\{\mathbf{m}_{sem}\neq 0\}} \tag{19}$$

## F.2. Hard Perturbation

Our framework operationalizes Pearl's theory (Pearl, 2009; 2012) of causal intervention by shifting the evaluation from the observational distribution $P(\mathbf{a}|\mathbf{V})$ to the interventional quantity $P(\mathbf{a} \mid do(\mathbf{V} := \mathcal{T}(\mathbf{V})))$, where the operator $do(\cdot)$ signifies an active manipulation of the data-generating process. By explicitly defining the transformation set $\mathcal{T} = \{\mathcal{T}_{tex}, \mathcal{T}_{geo}, \mathcal{T}_{patch}\}$ (as formulated in Eq. 20, 21, 22), we systematically intervene on specific environmental confounders: texture variation, geometric warping, and visual occlusion.

**Texture Perturbation.** To simulate complex surface variations and sensor noise without altering the underlying geometry, we employ a Fractional Brownian Motion (fBM) noise model. We generate the perturbation field $\mathcal{N}_{fbm}$ by aggregating 4

octaves of Gaussian noise, where each octave is spatially smoothed with a kernel $\sigma_k = 1.5 \cdot f_k$ (where frequency $f_k$ doubles per octave) and weighted by an amplitude decay of $0.5$. This multi-scale noise is normalized and added to the observation $\mathbf{V}$. The perturbation intensity is controlled by a scalar $\lambda \in [0, 1]$ and a base scaling factor $\alpha = 60.0$, ensuring the texture shift is visible but valid within the $[0, 255]$ pixel range. The operation is constrained to the specific semantic region defined by the binary mask $\mathbf{m}_{sem}$ (e.g., nuisance or support objects):

$$\mathbf{V}_{tex} = \text{clip}\left(\mathbf{V} + \lambda \cdot \alpha \cdot \frac{\mathcal{N}_{fbm}}{\text{std}(\mathcal{N}_{fbm})}, 0, 255\right) \odot \mathbf{m}_{sem} + \mathbf{V} \odot (\mathbf{1} - \mathbf{m}_{sem}) \tag{20}$$

**Geometric Perturbation.** To evaluate robustness against elastic deformations and camera lens distortions, we implement a structural warping mechanism based on random displacement fields. We sample independent horizontal and vertical flow fields $\delta_x, \delta_y \sim \mathcal{N}(0, 1)$ and smooth them using a Gaussian filter with $\sigma = 10.0$ to enforce local spatial continuity, simulating elastic material properties rather than white noise. The displacement magnitude is governed by $\lambda$ and a maximum pixel shift parameter $\beta = 25.0$. The perturbed image $\mathbf{V}_{geo}$ is reconstructed via bilinear interpolation at the displaced coordinates $\mathbf{p}' = \mathbf{p} + (\Delta x, \Delta y)$, applied exclusively within the target semantic region $\mathbf{m}_{sem}$:

$$\mathbf{V}_{geo}(\mathbf{p}) = \mathbf{V}(\mathbf{p} + \lambda \cdot \beta \cdot \hat{\delta}(\mathbf{p})) \odot \mathbf{m}_{sem}(\mathbf{p}) + \mathbf{V}(\mathbf{p}) \odot (1 - \mathbf{m}_{sem}(\mathbf{p})) \tag{21}$$

where $\hat{\delta}$ represents the normalized and smoothed displacement vectors.

**Patch Perturbation.** We employ randomized occlusion to simulate the loss of visual information or the presence of obscurants. Following the RISE algorithm utilized in our ISS computation, we generate binary occlusion masks $\mathbf{m}_{patch} \sim$ Bernoulli$(p)$ with a retention probability $p = 0.3$. These masks are sampled on a low-resolution $7 \times 7$ grid and bilinearly upsampled to the image resolution $(H, W)$ to create coherent blocking patches rather than scattered pixel noise. The perturbation replaces the original pixel values in the occluded regions with a blurred reference background $\mathbf{V}_{blur}$, thereby removing high-frequency information while preserving local color statistics:

$$\mathbf{V}_{patch} = \mathbf{V} \odot \mathbf{m}_{patch} + \mathbf{V}_{blur} \odot (\mathbf{1} - \mathbf{m}_{patch}) \tag{22}$$

| Task | Seed 0 | Seed 13 | Seed 50 | Seed 77 | Seed 99 | Mean $\pm$ Std |
|---|---|---|---|---|---|---|
| **Seen Tasks** | | | | | | |
| close_jar | 84 | 88 | 88 | 88 | 88 | $87.20 \pm 1.79$ |
| insert_onto_square_peg | 0 | 0 | 0 | 0 | 0 | $0.00 \pm 0.00$ |
| light_bulb_in | 52 | 56 | 52 | 52 | 52 | $52.80 \pm 1.79$ |
| meat_off_grill | 100 | 100 | 100 | 100 | 100 | $100.00 \pm 0.00$ |
| open_drawer | 84 | 80 | 84 | 80 | 84 | $82.40 \pm 2.19$ |
| place_cups | 0 | 0 | 0 | 0 | 0 | $0.00 \pm 0.00$ |
| place_shape_in_shape_sorter | 0 | 0 | 0 | 0 | 0 | $0.00 \pm 0.00$ |
| place_wine_at_rack_location | 100 | 100 | 100 | 100 | 100 | $100.00 \pm 0.00$ |
| put_groceries_in_cupboard | 28 | 28 | 28 | 28 | 28 | $28.00 \pm 0.00$ |
| put_item_in_drawer | 20 | 20 | 20 | 20 | 20 | $20.00 \pm 0.00$ |
| put_money_in_safe | 28 | 28 | 28 | 28 | 28 | $28.00 \pm 0.00$ |
| push_buttons | 100 | 100 | 100 | 100 | 100 | $100.00 \pm 0.00$ |
| reach_and_drag | 100 | 100 | 100 | 100 | 100 | $100.00 \pm 0.00$ |
| slide_block_to_color_target | 100 | 100 | 100 | 100 | 100 | $100.00 \pm 0.00$ |
| stack_blocks | 60 | 56 | 60 | 56 | 60 | $58.40 \pm 2.19$ |
| stack_cups | 20 | 20 | 20 | 20 | 20 | $20.00 \pm 0.00$ |
| sweep_to_dustpan_of_size | 100 | 96 | 100 | 100 | 100 | $99.20 \pm 1.79$ |
| turn_tap | 96 | 96 | 96 | 96 | 96 | $96.00 \pm 0.00$ |
| **Seen Average** | 59.56 | 58.44 | 59.78 | 58.89 | 58.67 | $59.07 \pm 0.58$ |
| **Unseen Tasks Level 1 (Similar Objects/Motions)** | | | | | | |
| close_fridge | 24 | 24 | 20 | 24 | 24 | $23.20 \pm 1.79$ |
| close_laptop_lid | 28 | 32 | 36 | 32 | 36 | $32.80 \pm 3.35$ |
| close_microwave | 52 | 48 | 40 | 40 | 44 | $44.80 \pm 4.97$ |
| lamp_off | 64 | 60 | 48 | 56 | 64 | $58.40 \pm 6.07$ |
| lamp_on | 52 | 52 | 60 | 48 | 40 | $50.40 \pm 7.70$ |
| open_grill | 8 | 12 | 16 | 8 | 0 | $8.80 \pm 6.10$ |
| phone_on_base | 52 | 60 | 68 | 60 | 52 | $58.40 \pm 6.57$ |
| put_books_on_bookshelf | 0 | 4 | 4 | 0 | 0 | $1.60 \pm 2.19$ |
| put_knife_on_chopping_board | 32 | 28 | 16 | 20 | 32 | $25.60 \pm 7.23$ |
| put_rubbish_in_bin | 28 | 20 | 16 | 20 | 12 | $19.20 \pm 6.76$ |
| put_toilet_roll_on_stand | 0 | 4 | 4 | 0 | 0 | $1.60 \pm 2.19$ |
| put_umbrella_in_umbrella_stand | 0 | 0 | 0 | 0 | 0 | $0.00 \pm 0.00$ |
| toilet_seat_down | 52 | 52 | 60 | 48 | 32 | $48.80 \pm 10.99$ |
| **Level 1 Average** | 30.15 | 30.46 | 29.85 | 27.38 | 25.85 | $28.74 \pm 1.97$ |
| **Unseen Tasks Level 2 (No Similar Objects/Motions)** | | | | | | |
| basketball_in_hoop | 12 | 12 | 8 | 8 | 8 | $9.60 \pm 2.19$ |
| beat_the_buzz | 4 | 4 | 0 | 4 | 4 | $3.20 \pm 1.79$ |
| scoop_with_spatula | 0 | 0 | 0 | 0 | 0 | $0.00 \pm 0.00$ |
| straighten_rope | 8 | 8 | 8 | 8 | 4 | $7.20 \pm 1.79$ |
| take_lid_off_saucepan | 32 | 20 | 16 | 16 | 12 | $19.20 \pm 8.29$ |
| take_plate_off_colored_dish_rack | 4 | 8 | 12 | 8 | 4 | $7.20 \pm 3.35$ |
| take_usb_out_of_computer | 100 | 100 | 96 | 96 | 100 | $98.40 \pm 2.19$ |
| turn_oven_on | 12 | 16 | 20 | 16 | 16 | $16.00 \pm 2.83$ |
| unplug_charger | 8 | 4 | 4 | 4 | 0 | $4.00 \pm 2.83$ |
| water_plants | 4 | 4 | 8 | 4 | 4 | $4.80 \pm 1.79$ |
| **Level 2 Average** | 18.40 | 17.60 | 17.20 | 16.40 | 15.20 | $16.96 \pm 1.21$ |
| **Total Unseen Average** | 25.04 | 26.26 | 24.35 | 23.13 | 21.22 | $24.00 \pm 1.92$ |

*Table 3.* Success rates (%) of $\pi_{0.5}$ on seen and unseen tasks.

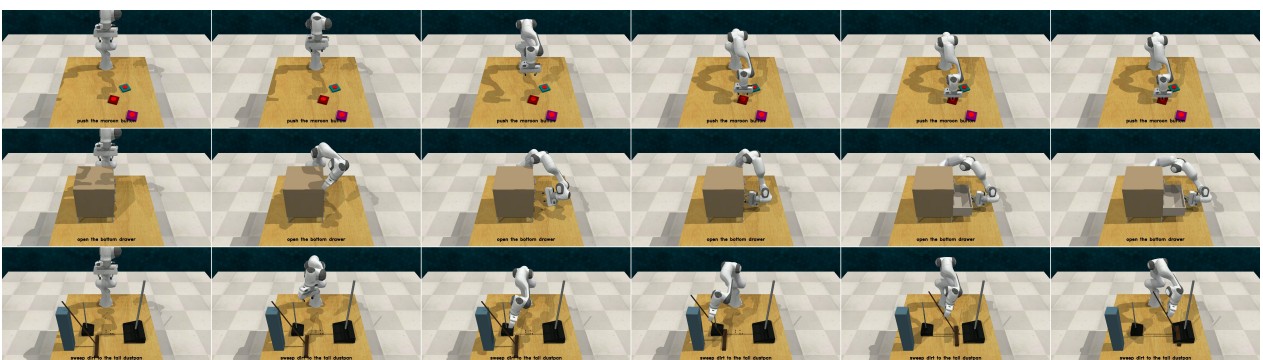

*(a)* **Successful Episodes of Seen Tasks:** "push the maroon button", "open the bottom drawer", "sweep dirt to the dustpan" (top to bottom).

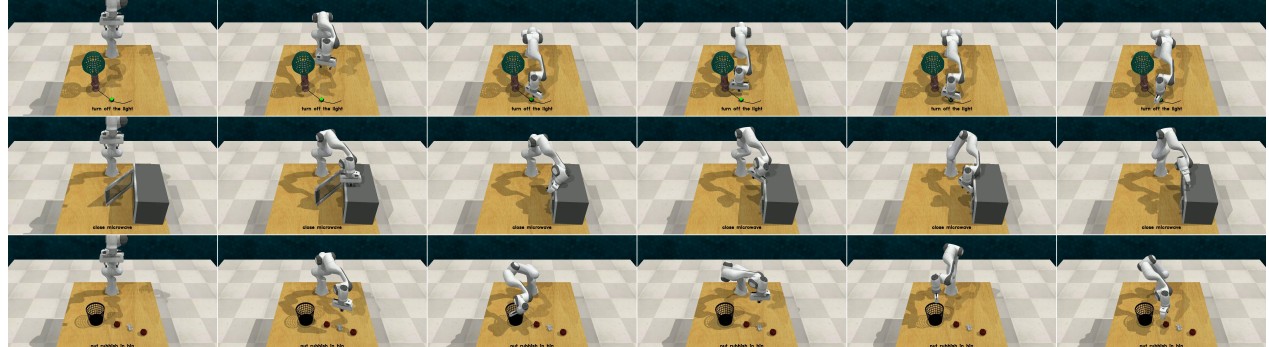

*(b)* **Successful Episodes of Unseen Tasks:** "turn off the light", "close microwave", "put rubbish in bin" (top to bottom).

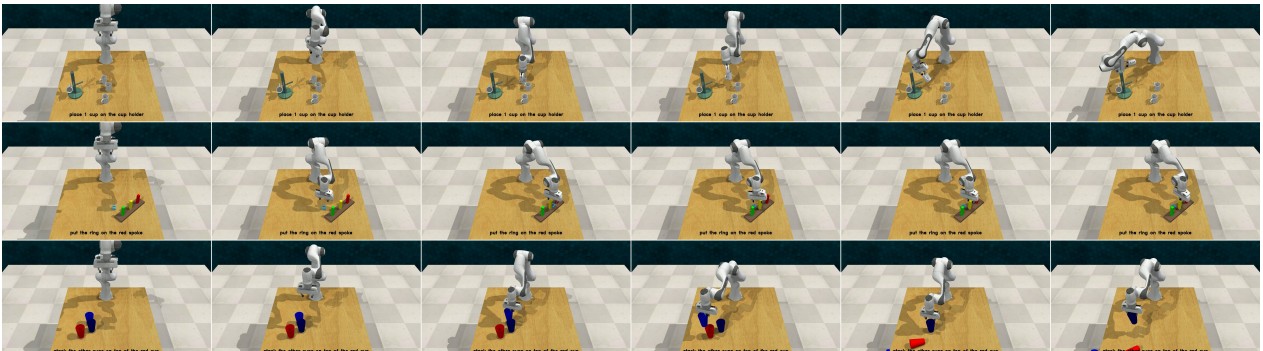

*(c)* **Failed Episodes of Seen Tasks:** "place 1 cup on the cup holder", "put the ring on the red spoke", "stack the other cups on top of the red cup" (top to bottom).

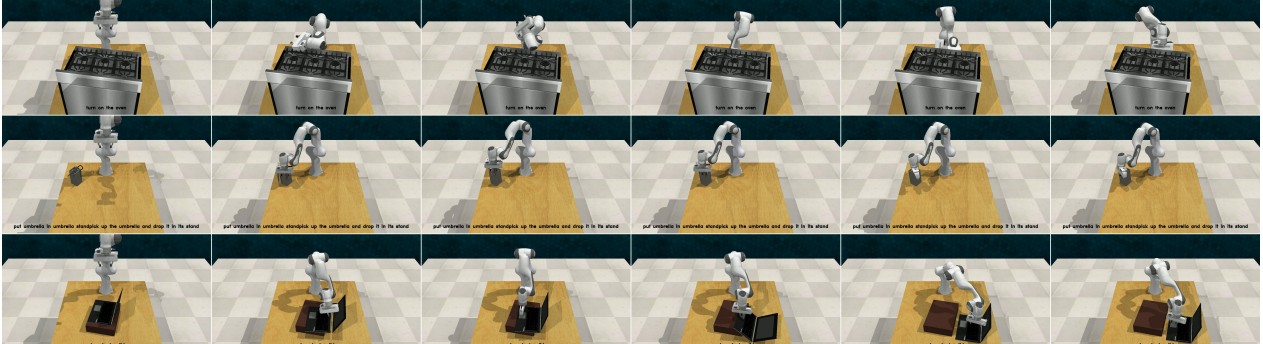

*(d)* **Failed Episodes of Unseen Tasks:** "turn on the oven", "put umbrella in umbrella stand", "close laptop lid" (top to bottom).

*Figure 13.* Representative episodes across seen and unseen tasks.

