# OpenReview forum: "Embodied Interpretability: Linking Causal Understanding to Generalization in Vision-Language-Action Models"
_ICML.cc/2026/Conference — ICML 2026 regular_

### Official Review · Reviewer_zKNU · 2026-02-27

**Soundness:** 3
**Presentation:** 2
**Significance:** 2
**Originality:** 2
**Overall Recommendation:** 4
**Confidence:** 2

**Summary:**

The paper confronts the problem that Vision-Language-Action models can be highly sensitive to spurious cues, e.g., in the images, and fail to generalize well. By formally treating visual-action attribution as an interventional estimation problem via do-calculus, the draft introduces two metrics: the Interventional Significance Score (ISS) and the Nuisance Mass Ratio (NMR). These mathematically and empirically quantify the degree to which a VLA policy relies on task-irrelevant visual features, establishing a correlation between spurious reliance and OOD generalization failure.

**Compliance With Llm Reviewing Policy:**

Affirmed.

**Final Justification:**

Given that the topic is so current and they appear to be first movers, I would say that it is worth having it at ICML, but I am no expert on the topic, and I think there could be more research done in this direction, going deeper into actual interpretability, which is why my confidence remains low, and the accept weak.

**Key Questions For Authors:**

1) it would be super interesting to get a bit more detail on the exact nature of these visual spurious cues. From Figure 1, it is hard to see what is the exact confounder (e.g. like the copyrighttags found in [1])
2) Why did you chose attributions and not counterfactuals? E.g. [2] found that it is quite hard to realize why something was highlighted by the attribution and often it is also practical to understand the "why?"
3) How can this insights gained now be used to actively improve my VLA model?




[1] Lapuschkin, Sebastian, et al. "Unmasking Clever Hans predictors and assessing what machines really learn." Nature communications 10.1 (2019): 1096.
[2] Bender, Sidney, et al. "Towards fixing clever-hans predictors with counterfactual knowledge distillation." Proceedings of the IEEE/CVF International Conference on Computer Vision. 2023.

**Limitations:**

From my understanding one has to know the "Visual Nuisance Regions", which limits applicability to a general scenario without such prior knowledge.

**Strengths And Weaknesses:**

Strengths:
+ highly relevant topic
+ interesting insight that problems known e.g. from classical image classification still persist in modern vision-language-action models


Weaknesses:
- not clear to me what spurious correlations are here exactly
- from my understanding one has to know the "Visual Nuisance Regions", which limits applicability
- that looking at attention maps for relevances provides bad results is an old finding
- i miss some actionability of the gained insight

---

> ### Author Rebuttal · Authors · 2026-03-30
>
> We thank the reviewer for acknowledging the relevance of the topic and the interesting insight of our paper, for highlighting the need for more concrete intuition, and for pointing out the Clever Hans literature.
>
> We agree that the Clever Hans literature is highly relevant, since both settings require attribution: apparent success may in fact rely on nuisance cues rather than true causal cues. The **key difference** is that, in Clever Hans, the attribution is already relatively clear: people could directly suspect that Hans was relying on the "trainer's body signals", and then verify through counterfactual tests by moving or altering the trainer. In VLA, by contrast, attribution is often unclear in advance and may vary over time, across camera views, and across task stages, because the policy produces sequential robot actions rather than a single output. Therefore, for VLA tasks, we must first identify which attributions truly influence the action at each moment and how they change over time. Our interventional attribution framework (ISS+NMR) is designed precisely for this purpose.
>
> **Key question 1.** *From Figure 1, it is hard to see what is the exact confounder.*
>
> We revise **Figure 1** following the presentation style of the suggested paper. The main changes are:
> 1. Add zoomed-in insets to highlight visual cues in the heatmap, with labels for visual cues in nuisance region (background, texture, shadow) and task-relevant cues (manipulator, end-effector, cups), following the region taxonomy (**Definition 4.2**).
> 2. Added segmentation mask overlays, coloring visual nuisance regions ($\Omega_{\text{nuis}}$) in red, action-critical regions ($\Omega_{\text{act}}$) in green, and environmental support regions ($\Omega_{\text{sup}}$) in blue.
> 3. Improve the heatmap color scheme to a higher-contrast gradient.
>
> The updated figure can be viewed at the anonymous link below.¹ We confirm that this link does not reveal any author or institutional information and is only meant to help reviewers see the revised figure without violating double-blind review.
>
> ¹ https://drive.proton.me/urls/YWRHR2NQ8G#Pd2XDSldiuSE
>
> **Key question 2.** *Why did you choose attributions and not counterfactuals?*
>
> The scope of our work is Interventional attribution rather than counterfactuals due to two reasons:
> 1. **They answer different questions.** Our goal is to identify **"what"** drives the VLA's current action, rather than **"why"** the action would change under a hypothetical input. For example, if the model picks up a mug, we want to know which visual regions drive that action, whether it is the mug itself, the background, or other regions we did not initially pay attention to.
> 2. **They require different assumptions.** ISS perturbs visual regions with Bernoulli masks and measures how the action changes (**Sec. 4.1**). This gives us a direct way to find important regions without defining alternatives in advance. By contrast, counterfactual analysis usually needs a predefined alternative target. In VLA tasks, however, this is often unclear, because the visual regions that matter can change across camera views and over time. Our "close jar" example (**Appendix D.2**) shows this clearly: in the front view, the model relies more on the robot arm, while in the wrist view, it relies more on the lid. The important regions can also change over time, shifting from the lid during pickup to the bottle during placement.
>
> We believe your concern is closely related to **Reviewer 1Sjf (Key Question 2)**, and our response there may also be helpful here. In short, that reviewer asked why we do not test the robustness of ISS by perturbing non-nuisance regions. The core point of our response is that perturbing task-relevant regions in VLA breaks task-related cues and makes action attribution unclear, so the resulting test becomes unreliable. Therefore, identifying task-relevant attributions and how they change is an essential problem in VLA research.
>
> **Key question 3.** *How can this insights gained now be used to actively improve my VLA model?*
>
> Our framework can be translated into a practical diagnostic tool for analyzing VLA reliance on visual nuisance regions and improving out-of-distribution (OOD) generalization, with the following uses:
> 1. Enabling quantitative comparison of OOD generalization across VLAs with NMR, while using ISS heatmaps for visual failure analysis.
> 2. Providing a training signal for improving VLA OOD generalization when integrated into training as an auxiliary supervision signal or regularization constraint.
> 3. Guiding the design of better multi-view or temporal fusion mechanisms by highlighting the most useful views and time steps.
> 4. Identifying misleading samples, views, or stages for data filtering, reweighting, or augmentation.
> 5. Supporting higher-quality data collection by identifying nuisance cues and reducing their influence during collection.

---

> > ### Author Rebuttal · Reviewer_zKNU · 2026-04-01
> >
> > Hi, thank you for the revised figure. I think that it is easier to understand already, and now I can see that shadows appear to be the confounder.
> >
> > However, I am not sure if I completely agree with everything you write. In image classification, it is also not clear what the confounders are beforehand. The point of explainable machine learning in general is to find out what problematic behauviour of the model is and understand it. I think I still miss a bit of interpretation there: Okay, the shadow is the confounder, but why? Is this generally a problem for all kinds of robots like this? Can we learn something more general about VLAs from this that people could look for when developing them?
> >
> > Moreover, the "what drives the action" and the "why not another action" are closely related, if not even the same question, to me in the end. The real difference between a counterfactual and an attribution is that an attribution tells you quite directly and straightforwardly the "where does my model look?" and a counterfactual would tell you "why does it look there?". I see, of course, that calculating a counterfactual would be more complicated and requires additional decisions, so I think the deeper question I wanted to ask with this is: Why do you assume that localization only suffices for your problem? Couldn't there also be color, texture, or size shifts when working with a real robot, which would be highly relevant and cannot be detected with an attribution?

---

> > > ### Author Response · Authors · 2026-04-01
> > >
> > > Thank you for the response and clarification. We understand your question better now.
> > >
> > > We want to clarify that our research goal focuses on **predicting VLA generalization**, not explainability itself. We use interventional attribution to identify what affects actions, and test whether this is sufficient to predict generalization under distribution shifts. We mainly want to validate: **if we use attribution as such a tool, does it work?** Our results show that it does (Pearson r=-0.77 between NMR and success rate).  Currently, there is no explanation tool that directly predicts VLA generalization. For example, as we cited in our submission, the benchmark [1] assesses VLA generalization using success rate, requiring rollouts across all test scenarios. Our tool provides an elegant solution. We fully agree that counterfactual can better answer "why", and as you noted, they need careful design. However, for predicting VLA generalization, we just want to test whether attribution is an effective way to do it, not to study which method provides more complete interpretability.
> > >
> > > Your comments are very helpful, and have made us realize that counterfactual could be valuable for future work on VLA interpretability. If possible, we would be very excited to discuss VLA interpretability with you.
> > >
> > > [1] Zhou, Jiaming, et al. "Exploring the Limits of Vision-Language-Action Manipulation in Cross-task Generalization." _The Thirty-ninth Annual Conference on Neural Information Processing Systems_.

---

### Official Review · Reviewer_HGG7 · 2026-03-08

**Soundness:** 3
**Presentation:** 2
**Significance:** 3
**Originality:** 3
**Overall Recommendation:** 4
**Confidence:** 2

**Summary:**

The paper focuses on improving the poor out-of-distribution (OOD) performance of Vision-Language-Action (VLA) policies. The authors propose a novel interpretability approach by introducing two measures: the Interventional Significance Score (ISS), which estimates the causal influence of visual regions on action decisions, and the Nuisance Mass Ratio (NMR), responsible for quantifying attribution to task-irrelevant features. The proposed framework was evaluated on the AGNOSTOS benchmark, demonstrating that NMR predicts policy generalization under distribution shifts, and ISS produces more consistent attributions than existing interpretability methods.

**Compliance With Llm Reviewing Policy:**

Affirmed.

**Final Justification:**

I update my recommendation to weak accept, with relatively low confidence.

The paper is well written, with a strong theoretical foundation and clear supporting visualisations and explanations. It addresses a timely and relevant problem, and the rebuttal adequately clarified concerns regarding practical impact and applicability.

The main remaining limitation is that the proposed framework is primarily a post-hoc diagnostic tool, which somewhat constrains the perceived scope of the contribution and could benefit from additional empirical validation, particularly in demonstrating how the diagnostic insights translate into improved training or decision-making.

Overall, despite this limitation, I find the work sufficiently strong to support acceptance, albeit with low confidence.

**Key Questions For Authors:**

While the two measures are described in detail, could the authors briefly summarise how they together translate into a ready-to-use framework for practitioners (e.g., recommended steps, usage guidelines, and interpretation of the results)? Additionally, it would be helpful to understand the potential practical impact of this framework, including its limitations.

**Limitations:**

yes

**Strengths And Weaknesses:**

Strengths:
- Strong theoretical foundation supported by clear mathematical notation.
- Detailed descriptions of the proposed algorithms.
- Numerous visualisations and diagrams that help illustrate and clarify the proposed measures.

Weaknesses:
- Roughly three pages of the main text are dedicated to diagrams and visualisations. While these are helpful, and some theoretical details are deferred to the appendices, the overall balance feels slightly uneven, with some potentially important explanations missing from the main text.
- In particular, the exact structure of the proposed framework is not entirely clear. Although two measures are presented in detail, additional clarification on how they should be applied in practice would be beneficial. Some practical nuances are currently underexplained, making it harder to understand how the components form a cohesive framework.
- Discussion and/or limitations section is missing.

---

> ### Author Rebuttal · Authors · 2026-03-30
>
> We thank the reviewer for the constructive feedback and for recognizing the theoretical foundation.
>
> The current submission does include a limitations discussion in the **Appendix G**, but we agree that it was not sufficiently signposted from the main paper. We also agree that a clearer discussion of the practical impact would be valuable, especially since **Reviewer zKNU** raised the similar point. If accepted, we will use the additional camera-ready page permitted by ICML to bring these points into the main text more explicitly.
>
> **Key question 1. (concern 1)**  *Could the authors briefly summarize how the framework translates into a ready-to-use framework for practitioners?*
>
> Our framework can be translated into a **post-hoc diagnostic tool** for assessing how well a VLA generalizes under distribution shift. Given any set of rollout episodes, whether from training or from a new environment, users can use this tool offline in two steps:
> 1. inspect **ISS** heatmaps to identify which visual regions influence the predicted action;
> 2. compute **nmr@10** to measure how much of the action attribution falls on nuisance visual regions.
>
> The interpretation is straightforward: If the heatmap highlights task-related regions, this suggests meaningful reliance; if it highlights nuisance regions, this suggests spurious reliance and weaker generalization. This is consistent with our empirical findings, where lower NMR correlates with better generalization under distribution shift (**Fig. 3**; **Appendix D.1**). For example, in the "close jar" case (**Appendix D.2**): the wrist view highlights the gripper fingers and the target jar lid, while the front view highlights the robot manipulator, illustrating how ISS reveals the distinct role of each view in the final prediction.
>
> **Key question 1. (concern 2)** *Additionally, it would be helpful to understand the potential practical impact of this framework, including its limitations.*
>
> This framework is used to test whether a VLA can be generalize under distribution shift. One core challenge in VLA is that its internal decision process is a black box, making failure analysis under distribution shift difficult (**Sec. 2**). Our framework provides a theory to address this challenge by detecting spurious vision-action causal structures inside VLA policies. In practice, this theory can be translated into a diagnostic tool: ISS provides qualitative heatmaps for visual regions influencing the action, while NMR provides a quantitative score of nuisance reliance. As a result, causal structures that were previously difficult to use can now be more directly leveraged to analyze and improve VLA models, as follows:
> 1. It enables researchers to quantitatively compare OOD generalization across VLAs with NMR and inspect ISS heatmaps for visual failure analysis.
> 2. It provides a training signal for improving VLA OOD generalization when integrated into training as a supervision signal or regularization constraint.
> 3. It guides the design of better multi-view or temporal fusion mechanisms by highlighting the most useful views and time steps.
> 4. It identifies misleading samples, views, or stages for data filtering, reweighting, or augmentation.
> 5. It supports higher-quality data collection by identifying nuisance cues and reducing their influence during collection.
>
> The limitation (**Appendix G**) can be summarized as follow:
> 1. ISS assumes unimodal action predictions, so it may miss mode-specific effects when the policy has multiple valid actions.
> 2. ISS needs multiple forward passes per state, making it much slower than attention-based methods and limiting it to offline use.
> 3. NMR depends on good nuisance masks, which are easy to get in simulation but harder and more error-prone in the real world.

---

> > ### Author Rebuttal · Reviewer_HGG7 · 2026-04-03
> >
> > Thank you for your clarifications; my concerns have been addressed. I find the technical contribution to be solid. At the same time, the presentation could still be improved. While I cannot fully assess improvements to the presentation in the rebuttal, the authors’ responses provide concrete and convincing plans to address these issues in a revised version.

---

> > > ### Author Response · Authors · 2026-04-03
> > >
> > > Thank you for the acknowledgment. We are glad that our responses fully addressed your concerns, that you find the technical contribution to be solid, and that our responses provide concrete and convincing plans to address the presentation issues in a revised version. If you feel the paper is better aligned with an **Overall Recommendation** above "3: Weak reject,” we would be very grateful if you would consider updating your rating accordingly.

---

### Official Review · Reviewer_1Sjf · 2026-03-11

**Soundness:** 3
**Presentation:** 2
**Significance:** 2
**Originality:** 2
**Overall Recommendation:** 4
**Confidence:** 1

**Summary:**

The paper introduces a causal testing method to check if vision language action models focus on the visual regions of interest.

**Compliance With Llm Reviewing Policy:**

Affirmed.

**Final Justification:**

In general, the idea you put forward in the paper seems to me relevant and timely. The authors also addressed all of my questions in their rebuttal. However, due to my lack of experience with the area of research and thus the methods and models discussed in the paper, I do not feel I can properly judge the Significance and Originality. I am therefore not comfortable raising my score above a weak accept.

**Key Questions For Authors:**

- In the Conclusion you write that NMR predicts generalization under distribution shifts. Can you explain in more detail how you show this? In general, the heatmaps to me look diffuse more than anything. Even for lower nuisance values the model concentrates saliency to points in the visual input thatt to me look very uninformative.
- In 5.2 you write that you are intervening by introducing noise or by explicitly modyfing for example texture, but always only in nuisane regions. Testing if manipulations in nuisance regions change the mean error over actions makes sense, but that to me seems like only one side of the medal. Shouldn't you also test if manipulating non-nuisance regions then leads to large error over actions?

**Limitations:**

There is an impact statement. However, authors do not discuss the limitations of their work in detail.

**Strengths And Weaknesses:**

I want to preface this by saying that I have not worked on vision language action models before and have no experience with the methods and models discussed in the paper. I am therefore not familiar with the related literature and I have not checked the mathematical details. I will thus give this review a low confidence and it should be taken into account with caution.

- **Soundness**: The paper seems to me technically sound as far as I can tell.
- **Presentation**: I found this paper hard to read and I wish for some results more intuition would be given (such as for example explaining how to read the heatmaps in Figure 6). In general, I feel the paper introduces a number of concepts and results without properly discussing them or putting them into context very well. For example, the conclusion mentions that NMR predicts policy generalization under distribution shifts, but I am not sure where concretely this is shown in the paper.
- **Significance and Originality**: I find it hard to judge the significance and originality given that I do not work in this area. However, in general the idea put forward by the paper seems to me relevant and timely.

---

> ### Author Rebuttal · Authors · 2026-03-30
>
> We thank the reviewer for recognizing the technical soundness, relevance, and timeliness of our work, and for pointing out where the presentation could be more intuitive.
>
> **Key question 1. (concern 1)** *In the Conclusion you write that NMR predicts generalization under distribution shifts. Can you explain in more detail how you show this?*
>
> We show this through the Pearson correlation between NMR and task success rate under distribution shift. Our benchmark is based on AGNOSTOS, which evaluates VLA generalization by training on seen tasks and testing on two unseen task sets, where generalization is only evaluated by task success rate and rollout videos (**Table 3; Figure 13**). We hypothesize that, if NMR is consistently correlated with task success rate across all training and evaluation results, then it can serve as an indicator of generalization under distribution shifts. Our results confirm this hypothesis: higher NMR is associated with lower task success, with the Pearson correlation reaching -0.77 (**Figures 3 and 10**). We admit that our conclusion omitted some necessary details, such as the correlation with generalization benchmark results, which may have caused some confusion. We will revise the conclusion in the camera-ready version to state: *“Empirically, NMR tracks policy generalization under distribution shifts, as indicated by its strong negative correlation with task success rate on AGNOSTOS.”*
>
> **Key question 1. (concern 2)** *In general, the heatmaps to me look diffuse more than anything. Even for lower nuisance values the model concentrates saliency to points in the visual input that to me look very uninformative.*
>
> Our goal is to faithfully reveal what truly influences the VLA's action decision. Such “diffuse” heatmaps suggest that VLA decisions may depend on nuisance cues rather than task-relevant causes. The heatmaps are easier to understand through their meaning. For example (**Figure 12**), Attention Score shows where the model focus on, while ISS shows which regions actually affect the decision. These are not always the same. VLA can focus on some regions without truly relying on it, like a student looking at the blackboard without really following the lecture. Finally, we recommend using NMR to quantify these results, rather than relying only on visual inspection.
>
> **Key question 2.** *In 5.2 you write that you are intervening by introducing noise or by explicitly modifying for example texture, but always only in nuisance regions. Testing if manipulations in nuisance regions change the mean error over actions makes sense, but that to me seems like only one side of the medal. Shouldn't you also test if manipulating non-nuisance regions then leads to large error over actions?*
>
> Our goal is to assess whether the VLA policy relies on spurious correlations or exhibits memorization. Therefore, interventions must preserve the core task semantics and should not alter the language instruction, the target object, or the robot itself. This is analogous to testing whether a student has truly understood a math problem or merely memorized its surface form: one should keep the underlying problem unchanged and vary only non-essential aspects such as wording or presentation, rather than replacing it with a different type of problem altogether. Otherwise, the observed difference would reflect a task change, not reliance on spurious cues. The same applies to VLA policies: if we perturb task-related regions, the visual task semantics are broken, and the VLA may produce very different actions. The resulting high action MSE becomes hard to interpret, because it may come from the semantic change itself rather than the intervention. Even if ΔISS and action MSE are strongly correlated under such conditions, this cannot reliably show that the explanation is robust and faithful. Therefore, in **Sec. 5.2**, we perturb only nuisance regions, which provides a cleaner test of whether the explanation is robust and faithful.
>
> We believe your concern is closely related to the point raised by **Reviewer zKNU (Key question 2)**, and our response there may also be helpful here. In short, that reviewer brought up the **"Clever Hans effect"**: Hans appeared to solve arithmetic problems, but in fact relied on the trainer’s body signals, and asked why we do not similarly use counterfactual analysis in VLA setting. The core point of our response is that both questions assume that we can specify the perturbation target in advance. In VLA, however, the action relevant visual cues are not fixed, but may vary over time, across camera views, and across task stages. Therefore, perturbing a predefined non-nuisance region may not provide a clean test of robustness or faithfulness, because it may also break task-relevant information.

---

> > ### Author Rebuttal · Reviewer_1Sjf · 2026-04-02
> >
> > Dear authors,
> >
> > Thank you for your answers, this clarifies the questions I had. In general, the idea you put forward seems to me relevant and timely. However, due to my lack of experience with the methods and models discussed in the paper, as well as the related work, I feel I can not properly judge the Significance and Originality of this work. Therefore, I am not comfortable raising my score above a weak accept at this point. I have alerted the AC to this and retain my low confidence value, in the trust that the AC will take my review into account accordingly.

---

> > > ### Author Response · Authors · 2026-04-03
> > >
> > > Thank you for the acknowledgment. We appreciate that you took the time to carefully review our paper, even though you reported that this is outside your main area of expertise. But we still believe your key questions are relevant, and we are glad that our responses fully addressed them. We appreciate your positive remark that our work is relevant and timely.

---

### Official Review · Reviewer_dzcX · 2026-03-15

**Soundness:** 4
**Presentation:** 2
**Significance:** 3
**Originality:** 3
**Overall Recommendation:** 4
**Confidence:** 3

**Summary:**

This paper investigates how VLA models often rely on spurious correlations and introduces two metrics to quantify this behavior.

The authors estimate the causal influence of visual regions by applying randomized Bernoulli masks to replace specific image tokens with blurred baselines and measuring the resulting action deviation, before calculating the overlap between the most influential regions and task-irrelevant background features. Experiments demonstrate that a higher reliance on background features negatively correlates with generalization performance, and the masking-based approach provides more reliable visual attributions than attention or norm baselines.

However, the framework is currently limited by its assumption of unimodal action distributions and a high computational overhead that restricts its application to offline evaluation.

**Compliance With Llm Reviewing Policy:**

Affirmed.

**Key Questions For Authors:**

1. Your framework relies heavily on ground-truth segmentation masks to compute the Nuisance Mass Ratio (NMR). How do you plan to adapt this metric for real-world robotic deployments where perfect segmentation is unavailable or noisy?
2. The theoretical justification for using action MSE as a proxy for KL divergence assumes a unimodal action distribution. How might this approach be adapted for state-of-the-art diffusion policies that inherently model multimodal action distributions?
3. Given the high computational overhead of Monte Carlo sampling for the masking interventions, is there a pathway to make the Interventional Significance Score (ISS) feasible for real-time, online monitoring? Maybe via amortized inference somehow?
4. The experiments currently validate the proposed metrics on a single VLA architecture. Do you anticipate that the strong negative correlation between NMR and generalization holds across fundamentally different architectures like OpenVLA or Octo?
5. Beyond using ISS and NMR as post-hoc diagnostic tools, have you explored using these causal attributions as a regularization signal during training? It would be interesting to see if penalizing high ISS scores on nuisance regions could actively improve out-of-distribution generalization.

**Limitations:**

Yes

**Strengths And Weaknesses:**

I found the connection between causal masking and generalization failure quite elegant, even if the paper feels a bit heavy on the notation in the early sections. It is always refreshing to see interpretability grounded in actual interventions rather than just highlighting attention maps.

## Strengths
1. The paper addresses a highly relevant problem in embodied AI by tackling the tendency of VLA models to overfit to spurious visual correlations rather than learning valid signals from observations.

2. The methodology is well-grounded in causal inference theory, utilizing approximations of do-calculus and Markov blankets to formally separate task-relevant features from environmental nuisances.

3. The authors provide a rigorous empirical evaluation, demonstrating that their masking-based metric correlates strongly with out-of-distribution task success and outperforms correlational baselines like attention weights and token norms.

4. The approach introduces a highly structured perturbation analysis, testing the fidelity of the visual attributions against geometric, textural, and noise-based shifts.

## Weaknesses
1. The framework heavily relies on ground-truth instance segmentation masks from a simulator to identify background nuisances, which is difficult and costly to replicate in real-world robotic deployments.
2. The reliance on Monte Carlo sampling for the randomized masking introduces a massive computational overhead, restricting the method strictly to offline, post-hoc analysis rather than real-time monitoring.
3. The theoretical justification assumes a unimodal action distribution by equating action mean squared error with Kullback-Leibler divergence, a proxy that may fail in highly ambiguous tasks or with multimodal architectures like diffusion policies.
4. The experimental scope is somewhat limited, as it validates the metrics on a single model architecture rather than exploring how these causal alignments differ across diverse embodied models.
5. In general, the figures & visualization are not very readable, I recommend in future revision & future papers, that the authors spend more time on designing figures with higher clarity.

---

> ### Author Rebuttal · Authors · 2026-03-30
>
> We thank the reviewer for the positive assessment and the detailed questions.
>
> **Key question 1.** *Your framework relies heavily on ground-truth segmentation masks to compute the Nuisance Mass Ratio (NMR). How do you plan to adapt this metric for real-world robotic deployments where perfect segmentation is unavailable or noisy?*
>
> As future work, we plan to explore SAM-series methods as a replacement for ground-truth masks, to relax this assumption in real-world settings. On the reliance on ground-truth segmentation, ISS itself does not require segmentation masks; the masks are needed to instantiate NMR as a quantitative nuisance diagnostic. In our experiments, simulation makes these masks easy to obtain, since the simulator camera can render semantic regions directly. In real-world deployment, one would instead rely on approximate instance segmentation or detector outputs, which would make NMR noisier but would not invalidate ISS itself. This dependence of NMR on nuisance-mask quality is a real limitation, and we state it explicitly in **Appendix G**.
>
> **Key question 2.** *The theoretical justification for using action MSE as a proxy for KL divergence assumes a unimodal action distribution. How might this approach be adapted for state-of-the-art diffusion policies that inherently model multimodal action distributions?*
>
> One possible extension is to model different action modes separately and score them independently, then analyze distribution changes caused by interventions at the mode level. On the theoretical use of action MSE as a proxy for KL divergence, we agree that the current justification is intentionally narrow. Our derivation assumes a fixed isotropic Gaussian surrogate, under which KL reduces to a scaled squared difference between predicted action means. We do not claim that this derivation automatically extends to diffusion or flow-matching policies in general. For multimodal policies, a distribution-aware divergence would be more appropriate than the current mean-based proxy.
>
> **Key question 3.** *Given the high computational overhead of Monte Carlo sampling for the masking interventions, is there a pathway to make the Interventional Significance Score (ISS) feasible for real-time, online monitoring? Maybe via amortized inference somehow?*
>
> One possible extension is to move heavy sampling offline and keep only lightweight scoring online. On computational overhead, we agree that the present formulation of ISS is inherently offline. Under the chosen configuration, the paper reports 5.18s latency and 0.19Hz throughput, and we therefore explicitly position ISS as a tool for offline safety verification and post hoc interpretability rather than real time monitoring. A learned or amortized approximation would be a meaningful extension, but it is outside the scope of the current paper.
>
> **Key question 4.** *The experiments currently validate the proposed metrics on a single VLA architecture. Do you anticipate that the strong negative correlation between NMR and generalization holds across fundamentally different architectures like OpenVLA or Octo?*
>
> We expect this correlation to hold across architectures, because ISS is model agnostic and can characterize the visual causal attribution underlying action decisions in a consistent way across different VLAs. At the same time, we agree that the current empirical validation is limited to one VLA architecture. Our claim is therefore not that the exact correlation value is universal, but that nuisance-heavy causal attribution is a useful diagnostic of causal misalignment. In the present paper, this relationship is strongly supported on the evaluated setup, where nmr@10 reaches (r=-0.77) and the same negative trend persists across other top-k thresholds in **Appendix D.1**. Large-scale cross-architecture validation is an important next step and is also noted in **Appendix G**.
>
> **Key question 5.** *Beyond using ISS and NMR as post-hoc diagnostic tools, have you explored using these causal attributions as a regularization signal during training? It would be interesting to see if penalizing high ISS scores on nuisance regions could actively improve out-of-distribution generalization.*
>
> On using ISS or NMR during training, we agree that this is a promising direction. In the current paper, we position them as post hoc diagnostics, and we have not yet used NMR as a training penalty, but this aligns well with our planned future work (**Appendix G**). Specifically, we are considering incorporating NMR as a regularization term or other form of supervisory signal to actively discourage reliance on nuisance regions during training.

---

> > ### Author Rebuttal · Reviewer_dzcX · 2026-04-03
> >
> > I think in the current form, I recommend that this work **should be accepted to ICML.**
> >
> > There are some important limitations with regard to the current work (regarding the practicality of the non-amortized estimator, and the assumption of unimodal action distributions -- especially given the recent work on flow & diffusion based policy models).

---

> > > ### Author Response · Authors · 2026-04-06
> > >
> > > Thank you for the thoughtful follow-up. We are encouraged that you recommend this work **should be accepted to ICML in the current form**. We believe future work needs to address these limitations highlighted in our response and paper (Appendix G), as they point to meaningful research directions.

---

### Decision · Program_Chairs · 2026-04-30

**Decision:**

Accept (regular)

**Comment:**

This paper describes an approach for visual action attribution in VLA policies as a problem of causal intervention. The paper introduces a new score and principled metrics for diagnosing whether policies rely on task-relevant versus spurious visual features. After rebuttal all four reviewers marked their concerns fully or partially resolved and converged on weak accept. There was uniform agreement that the topic of the paper is timely. The reviewers found the framework technically sound, well-motivated, and useful for diagnosing generalization failures in VLA policies. Thus, the AC recommends the paper to be accepted.